# HEB is required for the specification of fetal IL-17-producing γδ T cells

Tracy S.H. In[1,2], Ashton Trotman-Grant[1,2], Shawn Fahl[3], Edward L.Y. Chen[1,2], Payam Zarin[1,2], Amanda J. Moore[1,2], David L. Wiest[3], Juan Carlos Zúñiga-Pflücker[1,2] & Michele K. Anderson [1,2]

IL-17-producing γδ T (γδT17) cells are critical components of the innate immune system. However, the gene networks that control their development are unclear. Here we show that HEB (HeLa E-box binding protein, encoded by *Tcf12*) is required for the generation of a newly defined subset of fetal-derived CD73⁻ γδT17 cells. HEB is required in immature CD24⁺CD73⁻ γδ T cells for the expression of *Sox4*, *Sox13*, and *Rorc*, and these genes are repressed by acute expression of the HEB antagonist Id3. HEB-deficiency also affects mature CD73⁺ γδ T cells, which are defective in RORγt expression and IL-17 production. Additionally, the fetal TCRγ chain repertoire is altered, and peripheral Vγ4 γδ T cells are mostly restricted to the IFNγ-producing phenotype in HEB-deficient mice. Therefore, our work identifies HEB-dependent pathways for the development of CD73⁺ and CD73⁻ γδT17 cells, and provides mechanistic evidence for control of the γδT17 gene network by HEB.

[1] Sunnybrook Research Institute, 2075 Bayview Avenue, Toronto, ON M4N 3M5, Canada. [2] Department of Immunology, University of Toronto, 1 King's College Circle, Toronto, ON M5S 1A8, Canada. [3] Fox Chase Cancer Center, 333 Cottman Avenue, Philadelphia, PA 19111, USA. Correspondence and requests for materials should be addressed to M.K.A. (email: manderso@sri.utoronto.ca)

nterleukin-17 (IL-17)-producing γδ T (γδT17) cells are the major source of early IL-17 production in mucosal tissues, such as the lung, gut, and reproductive tract[1–4]. Although γδT17 cells are critical for clearing many bacterial and fungal infections[2,5,6], aberrant regulation of these cells influences the courses of many autoimmune diseases, including multiple sclerosis, type 1 diabetes, and psoriasis[7–9]. Furthermore, γδT17 cells can promote tumor progression[10–12], whereas interferon-γ (IFNγ)-producing γδ T (γδT1) cells have potent anti-tumor activity[13,14]. γδT17 cell programming involves high-mobility group (HMG) box family members, such as Sox13 and Sox4, and the transcription factor RORγt[15]. Furthermore, T-cell receptor (TCR) signal strength can

influence the γδT17 cell fate[16,17], and the E protein antagonist Id3 is upregulated in response to TCR signalling[18,19], suggesting a function for Id and E proteins in the γδT17/γδT1 cell fates. Here we investigate the role of the E protein HEB (HeLa E-box binding protein, encoded by *Tcf12*) in γδT17 cell development.

γδ T cells are functionally programmed in the thymus, unlike conventional αβ T cells that acquire effector function in the periphery[15,20–23]. Specific TCRγ chains are linked to distinct effector programs, but the nature of these links is not clear[23,24]. γδT1 cells are primarily CD27⁺CD44⁻ cells[21], and typically express Vγ5, Vγ1, or Vγ4 (nomenclature of Tonegawa and colleagues[25,26]). γδT17 cells are mostly CD27⁻CD44⁺ cells, and express either Vγ6 or Vγ4[27]. Additionally, some Vγ1⁺ cells are capable of producing IL-4 and IFNγ [28]. The generation of different types of γδ T cells occurs in successive waves during fetal thymic development. Vγ5⁺ γδ T cells appear first in the fetal thymus and migrate to the skin[29]. Vγ6⁺ γδ T cells arise next, and home to mucosal tissues and secondary lymphoid organs[4]. These two γδ T cell subsets are produced only in the fetal thymus, followed by lifelong residence in their respective tissues. Vγ4⁺ and Vγ1⁺ cells, which first appear in late fetal and neonatal life, continue to be generated throughout adult life[30].

Functional programming of γδ T cells occurs after surface expression of the γδ TCR, but prior to exit from the thymus[31]. One model posits that TCR signal strength instructs specification of γδ T cell effector fate choice, with a stronger signal inducing the γδT1 cell fate[20,32,33]. TCR signalling involves activation of the PKCθ pathway, leading to activation of NFκB; the PI3K pathway, leading to activation of NFAT; and the MAPK signalling pathway (in particular, ERK), which is upstream of many inducible transcription factors, including Egr3[34]. MAPK signalling and Egr3 activity have been linked to the αβ/γδ T cell[19] and γδT17/γδT1 cell lineage choices[17,35,36], in agreement with the signal strength model. However, other experiments have indicated that strong TCR signals are needed for the development of at least some γδT17 cells[16]. Thus, controversies remain regarding the function of TCR signalling in γδT17/γδT1 specification.

Id3 is one of the genes that is upregulated downstream of TCR-mediated ERK activity in developing thymocytes[18,19,37,38]. Id3 acts as an antagonist of the E protein transcription factors, and modulates their activity by forming inactive heterodimers[39]. Three main E protein transcription factors operate during T-cell development: E2A (E-box binding 2, encoded by *Tcf3*), HEBAlt (alternative form of HEB), and HEBCan (canonical form of HEB)[40–43]. E2A and HEB have very important functions in αβ T-cell development[42,44–47], but their functions in γδ T cell effector fate specification is unclear.

While TCR signals may negatively influence the γδT17 fate, Sox4 and Sox13 provide positive inputs into γδT17 cell development[15]. *Sox13*⁻/⁻ mice and *Sox4*⁻/⁻ mice lack γδT17 cells, and these genes are necessary for the upregulation of *Rorc*, which encodes RORγt. RORγt is required for the generation of all known IL-17-producing lymphocytes: αβ Th17 cells, γδT17 cells, and IL-17-producing type 3 innate lymphoid cells (ILC3)[48]. HEB has been implicated in direct control of the promoter of *Rorc* in Th17 cells[49], suggesting that it might also have a function in γδT17 cells. However, Stat3, Irf4, BATF, and Rorα, which are

essential for the generation of Th17 cells, are dispensable for γδT17 development[50–52]. Furthermore, Sox13 does not operate in αβ Th17 cells[15,53]. Given these two models, a critical question is whether E protein activity positively affects the Sox-RORγt network, within the context of developing γδ T cells, and whether Id3 activity can inhibit it.

Here we study mice with targeted deletions in the *Tcf12* locus to investigate a possible function for HEB factors in γδT17 development. We identify a new type of CD73⁻ HEB-dependent γδT17 cell subset that arises early in the fetal thymus, prior to the appearance of CD73⁺ γδT17 cells. Whereas CD73⁻ γδT17 cells are absent in the fetal thymus of HEB-deficient mice, CD73⁺ Vγ6⁺ cells are present. However, they are compromised in RORγt expression, and in their ability to make IL-17. We also show that Vγ4⁺ γδT17 cells, but not Vγ4⁺ γδT1 cells, are dependent on HEB. HEB can directly regulate *Sox4* and *Sox13*, and Id3 can inhibit expression of these transcription factors in γδ T cell precursors. These results place HEB at the top of the known gene network that regulates γδT17 cell programming, and provide a new link between TCR signalling and γδT17 cell fate determination.

## Results

**Identification of CD24⁻CD73⁻ γδT17 cells in the fetal thymus.** Most γδT17 cells are generated in the fetal thymus[22]. We therefore evaluated fetal thymic organ cultures (FTOCs) as a model system for studying γδ T-cell development. This approach allowed us to analyze precursors from germline HEB-deficient (HEB^ko) mice, which undergo embryonic lethality[46], and to capture successive waves of developing γδ T cells without the complications of migration. Development of γδ T cells in the adult thymus can be tracked using CD24 and CD73[32]. CD73 is upregulated in response to TCR signalling and marks commitment to the γδ T cell lineage, whereas CD24 downregulation indicates maturity (Fig. 1a)[32]. We performed a time course for CD24 and CD73 expression on all γδ TCR⁺ cells from FTOCs to assess the CD24/CD73 subset distribution in a fetal context (Fig. 1a). E14.5 fetal thymic lobes were placed in FTOCs and analyzed by flow cytometry at progressive time points. γδ T cells progressed from CD24⁺CD73⁻ (immature, uncommitted) to CD24⁺CD73⁺ (immature, committed) and CD24⁻CD73⁺ (mature, committed) populations over time, similar to the adult thymus[32]. However, CD24⁻CD73⁻ cells also appeared by day 4 (d4), and accumulated with time. We next analyzed ex vivo thymocyte populations from embryonic, neonatal, adult mice (Fig. 1b). CD24⁻CD73⁻ cells were apparent at E17.5, and peaked in frequency one day after birth. CD24⁻CD73⁻ cells declined during the first week of postnatal life and were nearly absent in the adult thymus. In the periphery, CD24⁻CD73⁻ cells were present in the spleen and lung, but not in the gut (Fig. 1g). Together, our results define a population of CD24⁻CD73⁻ γδ T cells that develop in the thymus during fetal and neonatal life, and persist afterwards in the adult periphery.

**γδ T cells develop along two different pathways.** We next examined expression of RORγt in the CD24/CD73 subsets by

**Fig. 1** Identification of a CD24⁻CD73⁻ γδT17 cell subset in WT thymus. **a, b** Schematic of γδ T cell differentiation in adult thymus, and developmental profile of γδ T cells in **a** E14.5 FTOC cultured for the indicated time points and **b** ex vivo thymocytes from E17.5 embryos, day 1 and day 7 neonatal mice, and weeks 4, 7, and 12 adult mice. **c** Frequencies of RORγt⁺ and RORγt⁻ cells among all γδ T cells in WT E17.5, day 1, and day 7 ex vivo thymus, and FTOC day 7 and 10. **d, e** Representative FACS plots of the CD24/CD73 profiles of RORγt⁺ **d** and RORγt⁻ **e** γδ T cells. **f** Quantification of the proportions of CD24/CD73 developmental subsets within RORγt⁺ vs. RORγt⁻ γδ T cells. **g** Representative FACS plots of CD24/CD73 subsets in spleen, lungs and gut of WT (HEB^fl/fl) mice. All plots are gated on CD3⁺TCRγδ⁺ cells. Numbers in FACS plots indicate frequency within each gate. Data are representative of at least two experiments with 3 mice per group. Center values indicate mean, error bars denote s.e.m. *p*-values were determined by two tailed Student's *t*-test. NS = not significant

intracellular flow cytometry (Fig. 1c). Nearly all CD24⁻CD73⁻ cells in the E17.5 and d1 thymus expressed RORγt whereas CD24⁻CD73⁺ cells did not (Fig. 1d–f). The CD24⁺CD73⁻ cells displayed heterogeneity in RORγt expression at all-time points. At E17.5 and d1, CD24⁺CD73⁺ cells were negative for RORγt, but RORγt⁺ CD73⁺ cells were present by d7. We next stimulated ex vivo thymocytes with PMA/ionomycin and analyzed IL-17 production (Supplementary Fig. 1a). At E17.5, very few γδ T cells were able to produce IL-17, but at d1 post-birth, a robust population of CD73⁻ IL-17 producing γδ T cells was present, along with some CD73⁺ IL-17 producers. At d7, the frequencies of IL-17 producers had dropped considerably, but both CD73⁻ and CD73⁺ γδ T cells were contained within this population (Supplementary Fig. 1b). Thus, there may be two pathways of γδ T-cell development in fetal and neonatal life. Pathway 1 requires upregulation of CD73 prior to CD24 downregulation, resulting in a mature CD73⁺ population including both γδT17 and γδT1 cells. By contrast, γδ T cells differentiating along Pathway 2 mature directly into RORγt⁺ CD73⁻ cells by downregulating CD24 without overtly passing through a CD73⁺ intermediate, and generate only CD24⁻CD73⁻ γδT17 cells.

**CD24⁻CD73⁻ γδ T cells develop without upregulating CD73.** In the adult thymus, both Vγ4⁺ and Vγ1⁺ cells transit through a CD24⁺CD73⁺ stage, with very similar gene expression profiles[54]. To define the developmental pathway leading to CD24⁻CD73⁻ γδ T cells, we performed a precursor-product experiment. CD24⁺CD73⁻, CD24⁺CD73⁺, and CD24⁻CD73⁺ cells were sorted from E14.5 d3 FTOCs (Fig. 2a), and placed into host TCRδ⁻/⁻ fetal lobes that had been treated with deoxyguanosine for 2 days. This strategy ensured that the γδ T cell products were donor-derived. Analysis of FTOCs by flow cytometry clearly showed that only CD24⁺CD73⁻ precursors gave rise to CD24⁻CD73⁻ γδ T cells (Fig. 2b, c), confirming Pathway 2 (Fig. 2d).

**Vγ chain repertoire segregates with CD73 expression.** γδ T cells bearing different Vγ chains develop in successive waves during ontogeny, and effector function has been linked to Vγ chain usage. Therefore, we analyzed the Vγ chain repertoire in ex vivo thymocytes (Fig. 3a, b). We focused our analysis on the γδT17 cell-associated Vγ4 and Vγ6 chains, and examined Vγ1⁺ and Vγ5⁺ cells together. At E17.5, Vγ6⁺ cells were transiting through both Pathway 1 and Pathway 2, with populations of CD24⁺CD73⁺ cells and CD24⁻CD73⁻ cells present, whereas at d1, Vγ6⁺ CD73⁻ cells predominated. At E17.5, all Vγ4⁺ cells were immature (CD24⁺CD73⁻), but by d7 post-birth, mature Vγ4⁺ cells were present, mostly in the CD24⁻CD73⁻ subset. Vγ6⁺ cells are restricted to the fetal thymus[30] and thus were not assessed at d7. Similar patterns of Vγ chain usage and CD73 expression were observed in a time course of FTOC development (Supplementary Fig. 2). Thus, both Vγ6⁺ and Vγ4⁺ cells can use Pathway 2, but at different times during ontogeny. The Vγ1/5⁺ population, by contrast, contains very few mature CD73⁻ cells at any time point, and are mostly restricted to Pathway 1.

**RORγt is heterogeneous in all CD24⁺CD73⁻ γδ T cells.** We next evaluated the expression of RORγt in each Vγ chain subset in E17.5 γδ T cells (Fig. 3c). Within the Vγ6⁺ subset, RORγt⁺ cells were immature CD24⁺CD73⁻ cells or mature CD24⁻CD73⁻ cells (Pathway 2). By contrast, the RORγt⁻ Vγ6⁺ cells consisted of CD24⁺CD73⁻ cells and CD24⁺CD73⁺ cells (Pathway 1). Interestingly, the Vγ1/5⁺ γδ T cell subset, which does not generally adopt a γδT17 cell fate, also contained RORγt⁺ cells; however, these were exclusively immature. Therefore, RORγt was expressed in a heterogeneous manner in all CD24⁺CD73⁻ γδ T cells,

independently of Vγ chain usage, at E17.5. However, once cells exited the CD24⁺CD73⁻ stage, Vγ6⁺ cells could develop along Pathway 2, whereas Vγ1/5⁺ cells could not.

**γδT17-associated genes are expressed in CD73⁻ γδ T cells.** We performed qRT-PCR on γδ T cell subsets sorted from d7 FTOCs to evaluate a possible function for HEB in γδT17 cell development (Supplementary Fig. 3). *Sox13* and *Sox4* were highly expressed at the CD24⁺CD73⁻ stage, relative to the other developmental subsets, consistent with their expression in the adult thymus[32]. *Rorc* and *Il17a* were also expressed in this subset, at relatively low levels, and at higher levels in CD24⁻CD73⁻ cells. Pathway 1 progression (CD24⁺CD73⁻ to CD24⁺CD73⁺ to CD24⁻CD73⁺) was accompanied by *Sox13*, *Sox4*, HEBAlt, and HEBCan downregulation, the loss of *Rorc*, and the upregulation of *Egr3* and *Tbx21* (T-bet). By contrast, Pathway 2 (CD24⁺CD73⁻ to CD24⁻CD73⁻) resulted in upregulation of *Rorc*, lowering of *Sox13*, HEBAlt and HEBCan, and loss of *Sox4*. *Id3* was highest in CD24⁺CD73⁻ cells and CD24⁺CD73⁺ cells. It decreased in all mature γδ T cells, but had lower levels in CD24⁻CD73⁻ cells than in CD24⁻CD73⁺ cells. Therefore, HEB and γδT17-associated gene expression were correlated, whereas Id3 was less tightly associated with specific subsets, at least at the population level.

**γδ T cells develop in HEBᵏᵒ FTOCs.** The similarities between *Sox13* and HEB expression suggested a potential function for HEB in γδT17 development. We assessed this possibility by analyzing HEBᵏᵒ FTOCs. WT and HEBᵏᵒ embryos were obtained from timed-mated HEB heterozygous mice, and thymic lobes from E14.5 embryos were placed in FTOC for 7 days. As expected, HEBᵏᵒ FTOCs lacked double positive (CD4⁺CD8⁺) thymocytes, indicative of a severe block in αβ T cell development (Supplementary Fig. 4a), accompanied by a decrease in thymic cellularity (Supplementary Fig. 4d)[42]. The percentage of mature αβ T cells among all CD3⁺ T cells decreased, with a concurrent increase γδ T cells percentages, in the HEBᵏᵒ vs. WT FTOCs (Supplementary Fig. 4b, c). The total number of γδ T cells in HEBᵏᵒ FTOCs was about twofold less than in WT FTOCs (Supplementary Fig. 4d), consistent with earlier E18 ex vivo studies in the 129/B6 strain of HEBᵏᵒ mice[42].

**HEB is required for the generation of CD24⁻CD73⁻ γδT17s.** We next analyzed the CD24/CD73 γδ T cell subsets in WT and HEBᵏᵒ FTOCs. Strikingly, the CD24⁻CD73⁻ subset was nearly absent in HEBᵏᵒ cultures, at both d7 and d10 (Fig. 4a, b), consistent with a loss, rather than a delay, of the appearance of these cells. At both d7 and d10, the HEBᵏᵒ FTOCs contained CD73⁺ RORγt⁺ cells, consistent with an intact Pathway 1 (Fig. 4c, d). Similar proportions of WT and HEBᵏᵒ CD24⁻CD73⁺ cells were RORγt⁺ at d7, but there were fewer RORγt⁺ cells among the CD24⁻CD73⁺ cells in HEBᵏᵒ FTOCs at d10. We found a similar phenotype in ex vivo analysis of E17.5 WT and HEBᵏᵒ thymocytes in terms of the CD24/CD73 profile (Supplementary Fig. 5a) and the distribution of RORγt⁺ cells among the mature CD73⁺ and CD73⁻ subsets (Supplementary Fig. 5b). Therefore, Pathway 1 was at least partially accessible to RORγt⁺ HEBᵏᵒ γδ T-cell progenitors, whereas Pathway 2 was not.

**Defect in IL-17 production in HEBᵏᵒ fetal γδ T cells.** To evaluate γδT17 cell functionality, fetal thymocytes from d7 or d10 FTOCs were stimulated with PMA/ionomycin, followed by intracellular staining for IL-17 (Fig. 4e, f). Mature HEBᵏᵒ γδ T cells were greatly impaired in IL-17 production relative to WT. HEBᵏᵒ γδ T cells were also poorly responsive to cytokine stimulation (IL-1β, IL-21 IL-23, and IL-7) (Supplementary

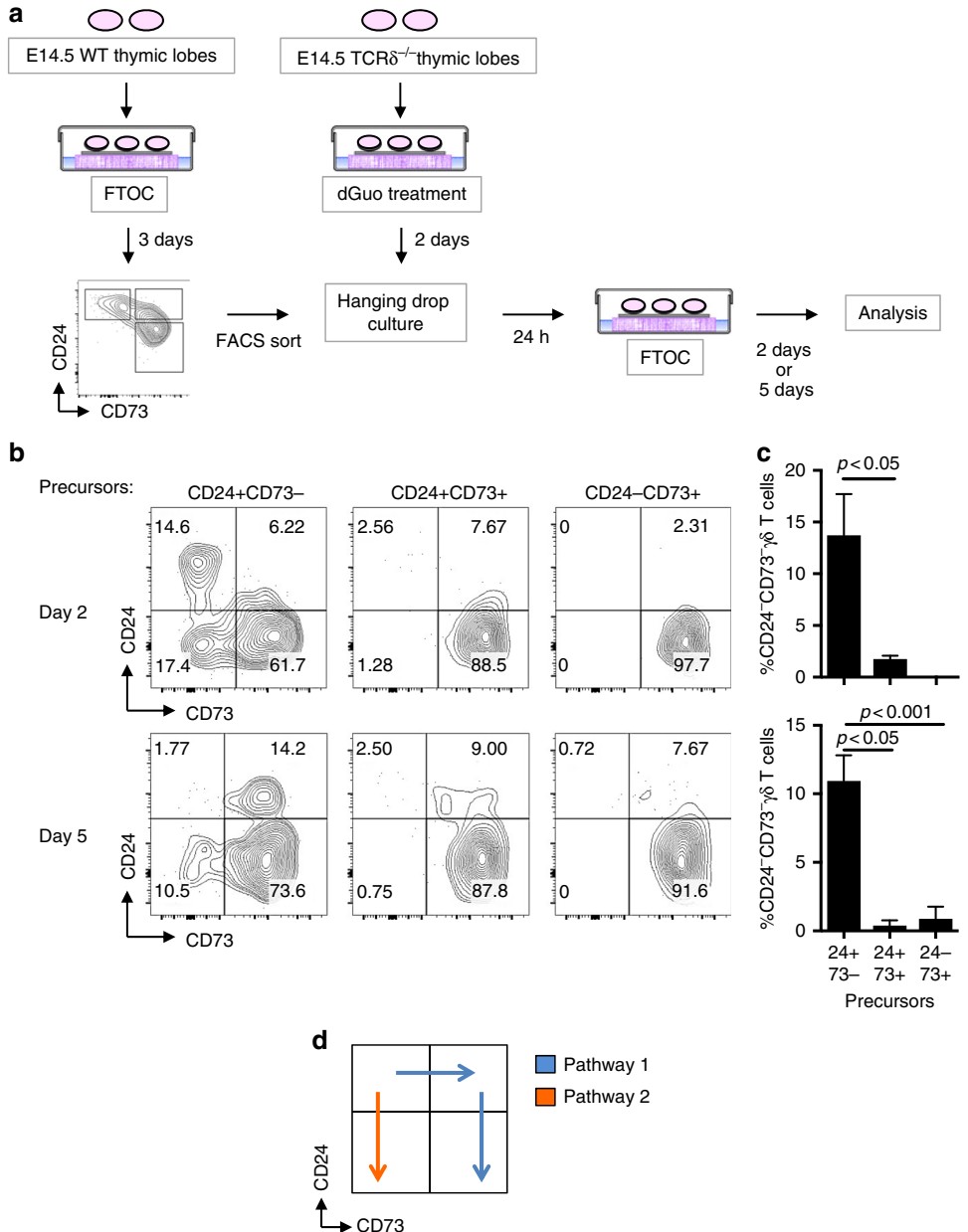

**Fig. 2** Developmental pathway of CD24⁻CD73⁻ γδT17 cells. **a** Schematic of the precursor-product experiment designed to determine the differentiation pathway of CD24⁻CD73⁻ γδT17 cells. Thymic lobes from E14.5 CD1 embryos were collected and cultured in FTOC for 3 days. Next, CD24⁺CD73⁻, CD24⁺CD73⁺, and CD24⁻CD73⁺ γδ T cells (all CD3⁺TCRγδ⁺) were sorted by FACS and incubated with E14.5 TCRδ-deficient thymic lobes overnight in hanging drop culture. The TCRδ-deficient lobes were prepared in advance by treatment with 2′-deoxyguanosine monohydrate (dGuo) for 2 days followed by resting for 1–2 days in media. The colonized lobes were transferred to FTOC and analyzed 2 and 5 days later. **b** Representative FACS plots of the developmental outcomes of the CD24⁺CD73⁻, CD24⁺CD73⁺, or CD24⁻CD73⁺ γδ T-cell precursors in FTOC for 2 or 5 days. Numbers in FACS plots indicate frequency within each gate. All plots are gated on CD3⁺TCRγδ⁺ cells. **c** Frequencies of CD24⁻CD73⁻ γδ T cells obtained from CD24⁺CD73⁻, CD24⁺CD73⁺, or CD24⁻CD73⁺ γδ T cells precursors at d2 and d5. **d** Schematic of the routes through development taken by γδ T-cell precursors on Pathway 1 (blue) vs. Pathway 2 (orange). Data are representative of at least three independent samples. Center values indicate mean, error bars denote s.e.m. $p$-values were determined by two tailed Student's $t$-test

Fig. 6a–c). By contrast, the percentage of IFNγ-producing γδ T cells was not significantly different between HEBᵏᵒ and WT (Supplementary Fig. 6b, right panel). ELISA for IL-17 production in WT and HEBᵏᵒ γδ T cells sorted from d7 FTOCs and stimulated with IL-1β, IL-21, IL-23, and IL-7 for 72 h (Supplementary Fig. 6d) confirmed a massive defect in the ability of HEBᵏᵒ γδ T cells to produce IL-17A, indicating a severe block in γδT17 cell programming in the absence of HEB.

**Sox4 and Sox13 mRNA expression are dependent on HEB.** To assess the effect of HEB on gene expression, we sorted the CD24/CD73 subsets from WT and HEBᵏᵒ d7 FTOCs, and analyzed mRNA levels by qRT-PCR (Fig. 5). CD24⁺CD73⁻ γδ T cells from HEBᵏᵒ FTOCs had lower levels of *Sox13* and *Sox4* expression than those from WT FTOCs, consistent with a role for HEB in γδT17 cell programming (Fig. 5a). By contrast, HEBᵏᵒ γδ T cells expressed levels of *Egr3* and *Tbx21* (T-bet) that were comparable to

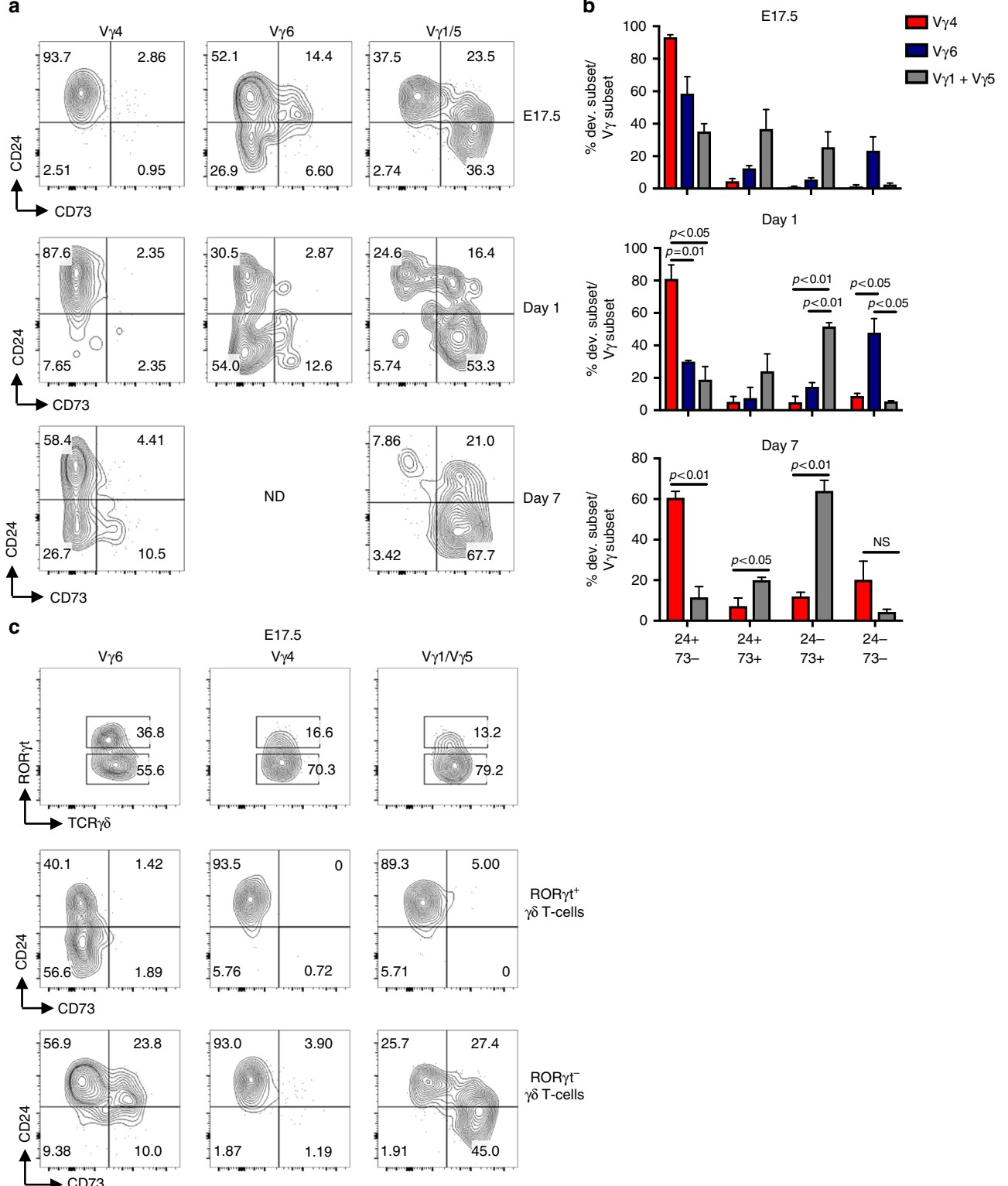

**Fig. 3** Developmental profiles of Vγ subsets. **a** Representative FACS plots of CD24/CD73 profiles of Vγ4, Vγ6, and Vγ1/5 cells in WT ex vivo thymus from mice aged E17.5, 1 day, and 7 days. **b** Proportions of C24/CD73 developmental subsets in each Vγ population (Vγ4, Vγ6, and Vγ1/5). For E17.5, all comparisons between each subset between Vγ populations are statistically significant with $p < 0.05$, except for the comparison of proportions of CD24$^-$CD73$^-$ cells within Vγ4 vs Vγ1/5. For day 1, comparisons are not statistically significant unless otherwise indicated. **c** RORγt$^+$ and RORγt$^-$ populations within Vγ subsets, and CD24/CD73 developmental profiles of RORγt$^+$ and RORγt$^-$ Vγ subsets. All plots are gated on CD3$^+$TCRγδ$^+$ cells. Numbers in FACS plots indicate frequency within each gate. Data are representative of at least three independent mice. Center values indicate mean, error bars denote s.e.m. $p$-values were determined by two tailed Student's $t$-test. ND = not done due to low cell number, NS = not significant

WT (Fig. 5b). Interestingly, *Id3* levels appeared to be lower in the CD24$^+$CD73$^-$ and CD24$^+$CD73$^+$ HEB$^{ko}$ immature subsets than in WT, although variability precluded statistical significance. Overall, our data indicate that γδT1 development is independent of HEB, whereas HEB is required for optimal *Sox* factor expression.

**Inhibition of *Sox4* and *Sox13* expression by Id3 overexpression.** *Sox4*, *Sox13* and HEB factors are all expressed in CD24$^+$CD73$^-$ γδ T cells, at the population level. If HEB directly regulates *Sox4* and/or *Sox13* in immature γδ T-cell precursors, then raising levels of Id3 would be expected cause a decrease in their expression.

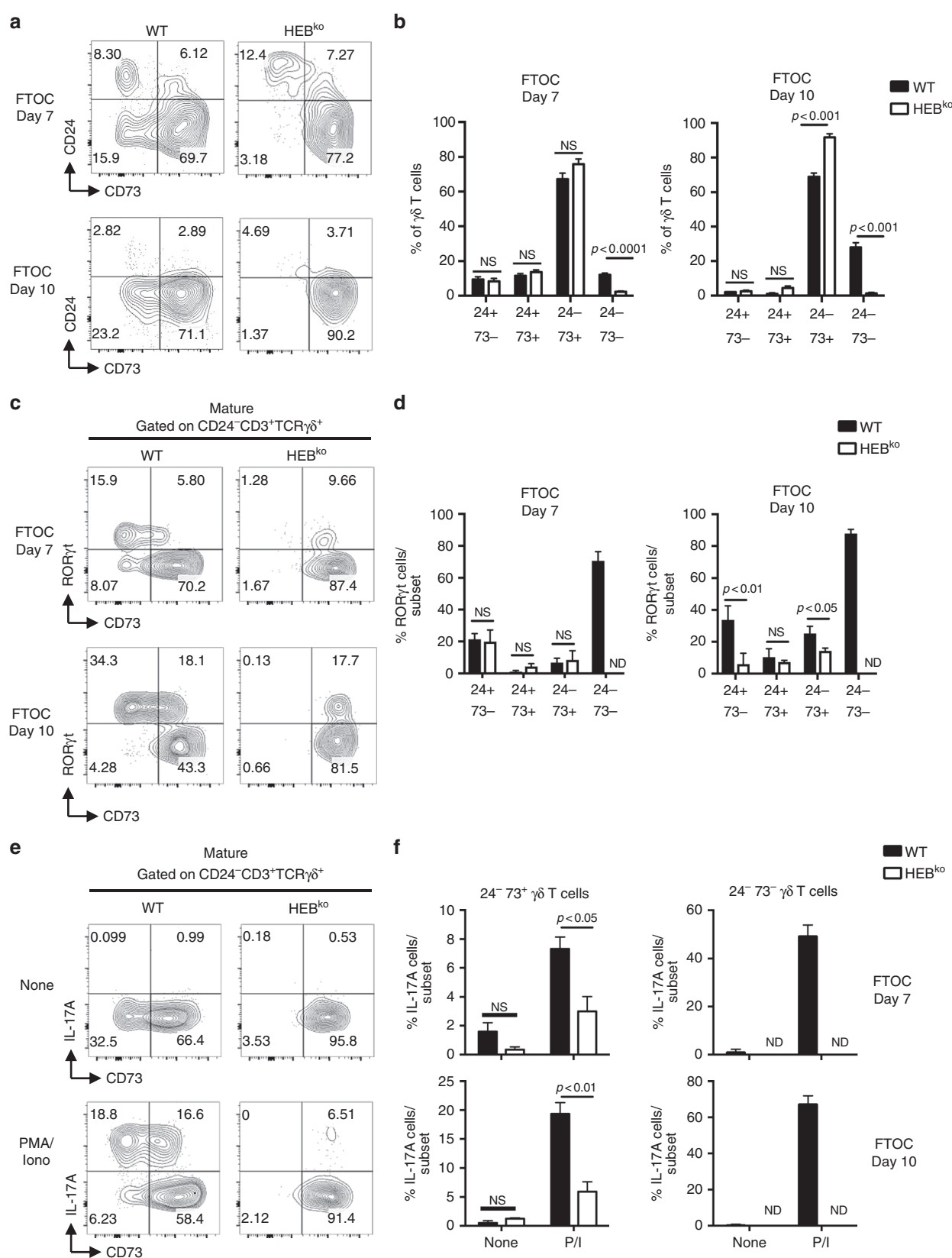

We generated T-cell precursors by culturing fetal liver precursors on OP9-DL4 cells, and transduced them with retroviral constructs expressing Id3/GFP or GFP only. After overnight culture, we sorted GFP+TCRγδ+CD3+CD24+CD73− cells and measured gene expression by qRT-PCR (Fig. 5c). *Id3* was clearly overexpressed in the Id3-transduced cells, whereas *Sox4*, *Sox13*, and *Rorc* were significantly decreased in the Id3-transduced cells, relative to the control. *Tbx21* (T-bet) was unchanged. The inhibition of *Sox* factor expression by acute Id3 expression in WT γδ T cells indicates that the decrease in their expression in HEB^ko γδ T cells was not due to upstream changes caused by germline HEB deficiency, or to differential cell death, and instead supports a direct function for HEB in regulating *Sox13* and *Sox4* expression in immature γδ T cells.

**HEB interacts with *Sox4* and *Sox13* regulatory region DNA.** Significant technical challenges are inherent in addressing the question of whether HEB directly regulates *Sox4* and *Sox13* in the context of developing γδ T cells, due to the very small numbers of TCRγδ+CD3+CD24+CD73− cells that can be obtained from fetal thymus. Therefore, we generated Rag-2-deficient DN3 cells, which provide the cellular context directly upstream of γδ T cell specification, using the OP9-DL4 system. Potential HEB binding sites were identified upstream of the transcriptional start sites of *Sox13* and *Sox4* (Fig. 6a, c), and ChIP was performed using anti-HEB antibodies. These experiments identified an HEB-occupied region ~25 kb 5′ of the *Sox4* transcriptional start site (Fig. 6b). A region 32 kb 5′ of the second start site of the *Sox13* locus contained 2 E-box binding sites (Fig. 6c), but was not detected by ChIP qRT-PCR. We noted that *Sox13* is upregulated significantly during the transition from the DN3 to the TCRγδ+CD3+CD24+CD73− stage, suggesting that DN3 cells might not have the right context to detect HEB binding to the *Sox13* locus. Therefore, we cloned this region into a luciferase reporter construct and tested its response to HEB co-expression by luciferase reporter assays. Strong activation was detected for both HEB proteins, especially HEBCan (Fig. 6d). Other regions of the *Sox13* locus containing E-box sites did not respond (data not shown). These results indicate that HEB can directly drive the expression of two key regulators of γδT17 fate determination.

**Alteration of the Vγ repertoire in the HEB^ko fetal thymus.** Previous reports have implicated E proteins in the direct regulation of some Vγ TCR genes[55,56]. We therefore examined Vγ chain in d7 FTOCs to assess whether the γδT17 cell deficiency in HEB^ko FTOCs could be due to the loss of γδT17-associated Vγ chain expression (Fig. 7). The frequency of Vγ4+ cells was greatly decreased in the HEB^ko fetal thymus compared to WT, whereas all other Vγ chains were present in similar proportions (Fig. 7a–c). Vγ6+ cells were inferred by the absence of Vγ1, Vγ4, and Vγ5, since there are very few Vγ7+ cells in the FTOCs (Fig. 7a, b). We also measured Vγ (*Vg*) chain mRNA expression by qRT-PCR using sorted γδ T cells from d7 FTOC (Fig. 7d), which confirmed the defect in Vγ4 expression and provided a

more direct measure of Vγ6 expression. Thus, HEB deficiency changes the fetal Vγ repertoire.

**Decrease in Vγ6+ γδT17 cells in HEB^ko thymus.** We next evaluated RORγt expression in WT and HEB^ko FTOCs within the mature Vγ4+ and Vγ6+ cells (Fig. 6e). Vγ4+ cells were examined only in the WT thymus due to their scarcity in the HEB^ko thymus. Very few mature Vγ4+ cells expressed RORγt. In contrast, WT Vγ6+ cells contained robust populations of both CD73− RORγt+ and CD73+ RORγt+ γδ T cells, especially in d10 FTOC. HEB^ko FTOCs lacked mature CD73− cells, as expected, but contained Vγ6+ CD73+ RORγt+ cells. The frequencies of RORγt+ cells among mature Vγ6+ CD73+ cells was indistinguishable between WT and HEB^ko FTOCs at d7, but was significantly lower in HEB^ko FTOCs at d10 (Fig. 6f). Therefore, the defect in IL-17A production in HEB^ko fetal γδ T cells is not merely due to the absence of the CD24−CD73− cells, or to the lack of Vγ4 cells. Instead, there appear to be two defects in the absence of HEB: (1) a lack of Vγ6 CD24−CD73− γδT17 cells, and (2) a lack of Vγ4 γδT17 cells, both of which follow Pathway 2 during normal fetal development.

**Postnatal appearance of Vγ4 cells in the HEB^cko thymus.** One possible explanation for the deficiency of Vγ4 cells in FTOC from HEB^ko mice would be the loss of incoming precursors that normally populate the thymus after E14.5. However, an ex vivo analysis of HEB^ko E17.5 thymus also showed a profound defect in Vγ4 cells, suggesting that this was not the case (Supplementary Fig. 7a). To further analyze the impact of HEB deficiency in neonatal and adult mice, we generated a mouse model lacking HEB in hematopoietic cells, by breeding HEB^fl/fl mice[57] with vav^Cre mice (HEB^cko mice). The phenotype of the HEB^cko FTOCs replicated the HEB^ko FTOC data (Supplementary Fig. 8). Therefore, the need for HEB in γδT17 development is intrinsic to the hematopoietic compartment. We next analyzed γδ T cells in the HEB^cko postnatal thymus, and found that Vγ4 cells were present, although in reduced proportions compared to WT, varying with age (Supplementary Fig. 7b). We also noted that all Vγ subsets were present in the spleen and lungs of HEB^cko mice, with the exception of Vγ5 cells, as expected[29]. As in the thymus, there was a reduction in the percentages of Vγ4 cells in these tissues (Supplementary Fig. 9).

**Defects in HEB^cko γδT17 cells in lung and spleen.** We next analyzed γδ T cell subsets in adult HEB^cko peripheral tissues. Unlike in the fetal thymus, some CD73− γδ T cells present in the spleen and lungs of HEB^cko mice. HEB^cko mice exhibited severe reductions in the proportions of Vγ4+ and Vγ4− cells expressing RORγt in the spleen (Fig. 8a) and the lungs (Fig. 8b), and the RORγt+ cells that were present were all CD73+. By contrast, the proportions of Vγ4+ cells expressing the γδT1 cell marker CD27 were increased (Fig. 8c). Interestingly, the frequency of CD27+ cells within the HEB^cko Vγ4− population was unchanged. This phenotype was accompanied by defects in the ability of peripheral

**Fig. 4** CD24−CD73− γδT17 cells do not develop in HEB^ko FTOCs. **a** Representative FACS plots of CD24/CD73 γδ T cell subsets in WT and HEB^ko FTOCs. **b** Quantification of the percentages of each CD24/CD73 developmental subset within all γδ T cells (CD3+TCRγδ+) in d7 and d10 FTOCs from WT and HEB^ko mice. **c** Representative FACS plots of thymocytes WT and HEB^ko FTOCs stained for intracellular RORγt and surface CD73 gated on the CD24− population. **d** Quantification of the frequencies of RORγt+ cells within the CD24/CD73 subsets in WT and HEB^ko FTOCs. **e** Representative FACS plots depicting intracellular IL-17A expression vs. CD73 expression in CD24− γδ T cells from WT and HEB^ko FTOCs after 5 h of stimulation with PMA/Ionomycin (PMA/Iono) and treatment with Brefeldin A. **f** Frequency of IL-17A+ cells within CD24−CD73+ or CD24−CD73−γδ T cells in FTOCs from WT and HEB^ko mice treated with Brefeldin A alone (None) or PMA/Iono and Brefeldin A (P/I) for 5 h. All plots are gated on CD3+TCRγδ+ cells. Numbers in FACS plots indicate frequency within each gate. Data are representative of at least three independent experiments with at least 3 mice per group. Center values indicate mean, error bars denote s.e.m. *p*-values were determined by two tailed Student's *t*-test. ND = not done due to low cell number. NS = not significant

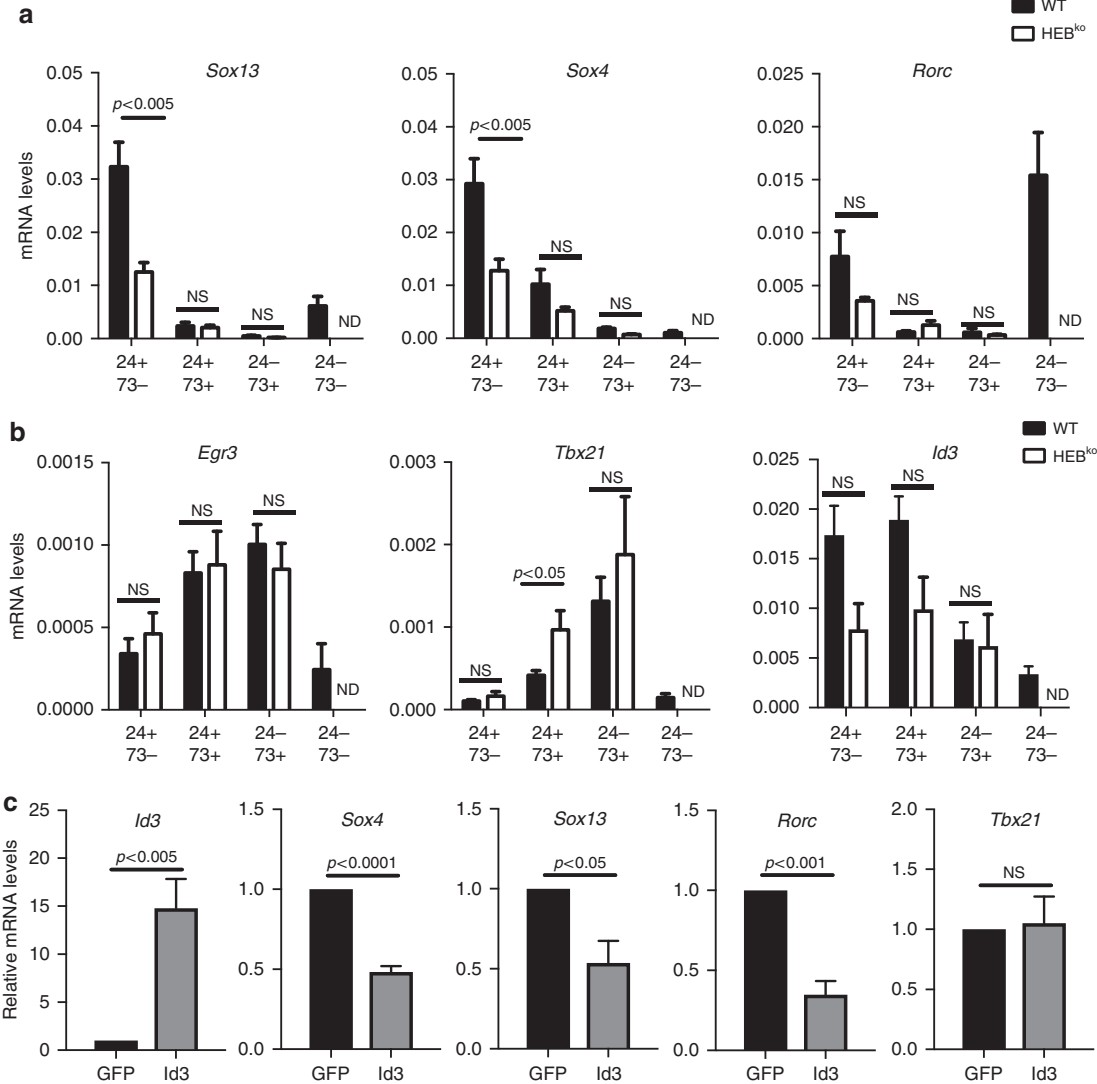

**Fig. 5** Decreased levels of γδT17 cell-related gene expression HEB$^{ko}$ γδ T cells. **a**, **b** qRT-PCR analysis of γδT17 cell-related genes **a** and γδT1 cell-related genes **b** in γδ T cell (CD3$^+$TCRγδ$^+$) subsets sorted based on the expression of CD24 and CD73 from WT and HEB$^{ko}$ FTOC day 7. Values are normalized to β-actin mRNA levels. Data are representative of at least three independent experiments with at least 3 mice per group. **c** qRT-PCR analysis of *Id3*, *Sox4*, and *Sox13* mRNA expression in CD24$^+$CD73$^-$ γδ T cells sorted from d3 FTOC-derived thymocytes that had been transduced overnight with GFP only, or GFP and Id3. Relative mRNA levels were calculated as the expression fold change over the GFP control. Data are representative of two independent experiments with 2 samples per group. Center values indicate mean, error bars denote s.e.m. *p*-values were determined by two tailed Student's *t*-test. ND= not done due to low cell number. NS= not significant

HEB$^{cko}$ γδ T cells to make IL-17, in response to either cytokines or PMA/ionomycin (Supplementary Fig. 10). Altogether, our results indicate critical functions for HEB in the specification of all γδT17 cells during multiple routes of development (Fig. 9).

## Discussion

γδT17s are critical mediators of barrier tissue immunity and autoimmune pathology. Here we have identified critical functions for HEB transcription factors in the generation of γδT17 cells. First, HEB is indispensable for the development of a fetal and neonatal wave of RORγt$^+$ CD24$^-$ CD73$^-$ γδ T cells (Pathway 2). Second, HEB is important for allowing a later fetal wave of CD73$^+$ cells to adopt a RORγt$^+$ γδT17 cell fate (Pathway 1). Third, HEB is required for the appearance of fetal Vγ4 γδ T cells, and adult Vγ4 γδT17 cells. All of these functions of HEB are linked through regulation of genes that are necessary for γδT17 cell programming. Taken together, our results clearly show that HEB is required for γδT17 cell generation through multiple developmental pathways,

and provide novel insights into the molecular networks that control γδT17 cell effector fate.

Our work has uncovered a distinct developmental pathway, Pathway 2, for γδT17 cell specification and maturation. Pathway 2 is restricted to Vγ6$^+$ and Vγ4$^+$ T-cell progenitors, gives rise only RORγt$^+$ γδ T cells, and occurs only during fetal and early neonatal development. It is yet to be determined how CD24$^+$CD73$^-$ cells are directed to enter one pathway vs. the other, but our results clearly show that HEB activity is required for entry of γδ T-cell precursors into Pathway 2. HEB-deficient γδ T cells are generated via Pathway 1, but they express very little RORγt, and are defective in IL-17 production. Therefore, HEB is required for the maturation of γδ T cells destined to become CD24$^-$CD73$^-$ γδT17 cells, in addition to its control of the gene network associated with IL-17 programming.

Our results suggest that the γδ T cell effector lineage choice occurs at the CD24$^+$CD73$^-$ stage of development. Controversy exists as to the relative importance of TCR signalling in driving the

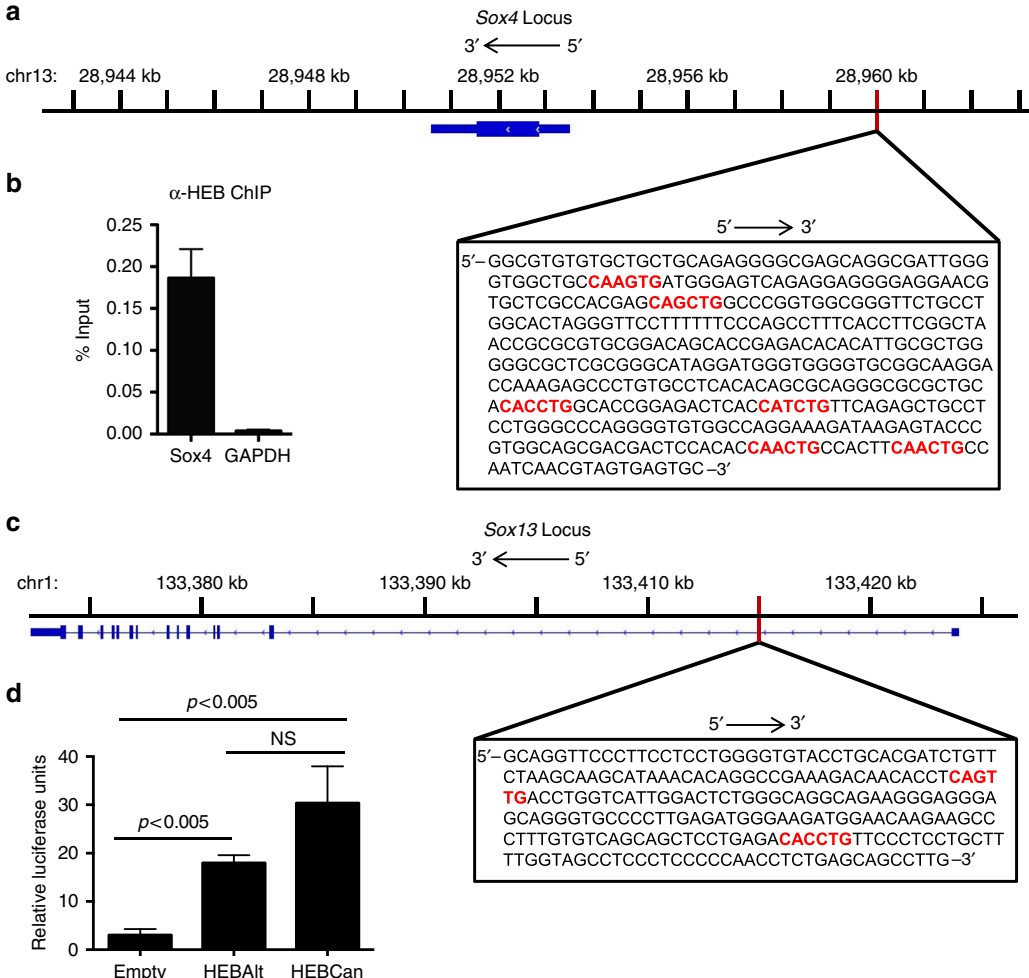

**Fig. 6** HEB factors interact with *Sox4* and *Sox13* regulatory regions. **a** Structure of the *Sox4* locus (oriented 3′–5′) and nucleotide sequence (boxed; oriented 5′–3′) of the region targeted for amplification to detect occupancy by HEB in Rag-2 deficient DN3 cells derived from fetal liver OP9-DL4 co-cultures. Blue boxes and lines indicate intron-exon structure of the *Sox4* locus. Potential HEB E-box binding sites are depicted in red. **b** quantitative real time-PCR results for detection of HEB or GAPDH protein occupancy on the region depicted in **a** after chromatin immunoprecipitation of lysates using anti-HEB antibodies. ChIP qRT-PCR results are shown as percent of input. Primers designed to amplify GAPDH were used as a negative control. **c** Structure of the *Sox13* locus (oriented 3′–5′) and nucleotide sequence (boxed; oriented 5′–3′) of the region targeted for amplification by PCR to generate a luciferase reporter construct. Blue boxes and lines indicate intron-exon structure of the *Sox13* locus. Potential HEB E-box binding sites are depicted in red. **d** Luciferase assay results from 293T cells transfected with the firefly luciferase reporter construct containing the *Sox13* region shown in **c**, the Renilla luciferase construct, and pCMV-based expression vectors (empty, HEBAlt, HEBCan). Results are representative of two independent experiments with three technical replicates each. Values are normalized to Renilla. Center values indicate mean, error bars denote s.e.m. *p*-values were determined by two tailed Student's *t*-test

γδT17 cell fate choice. Moreover, some studies indicate that intrinsic bias toward one fate or another may exist prior to TCR signalling[15,31]. The heterogeneity of RORγt in CD24⁺CD73⁻ γδ T-cell precursors suggests that regulators of *Rorc* expression, including HEB, Sox13, and Sox4, may be likewise partitioned into different cell populations. Very little RORγt is expressed in CD24⁺CD73⁺ cells. Therefore, either the RORγt⁻ fraction of the CD24⁺CD73⁻ population is specifically selected into Pathway 1, or RORγt is downregulated as CD73 is upregulated. Future experiments using fate mapping approaches will help to resolve these possibilities.

In the fetal thymus, most of the developing γδ T-cell precursors express Vγ5 or Vγ6, and the majority of the Vγ4 and Vγ1 bearing cells are confined to the CD24⁺CD73⁻ population. Only Vγ5⁺ cells enter Pathway 1 at this time in ontogeny. Vγ5⁺ cells are programmed for IFNγ production by engagement of *Skint1*[17,58], in a manner that is dependent on signalling through the ERK-Egr3 axis. Vγ5⁺ cells develop in *Skint1⁻/⁻* mice, but they produce

IL-17 rather than IFNγ. These results provide evidence that ERK and Egr3 inhibit γδT17 cell programming during Vγ5 development. Furthermore, strong γδ TCR signalling inhibits γδT17 cell programming in an ERK-dependent manner[36], and Id3 is more strongly induced in cells exposed to strong TCR signals than weak TCR signals, through activation of Egr3[19]. These results support a function for Id3 in repressing the γδT17 cell fate. Our work complements these studies by defining HEB as a positive regulator of the γδT17 cell network. The similarity of the effect of Id3 overexpression and loss of HEB on the expression of *Sox4* and *Sox13*, in the relevant physiological context, provides compelling evidence that HEB acts in uncommitted γδ T cells to drive specification toward the γδT17 cell fate, and that Id3 can interfere with this process. Therefore, our identification of HEB as a top tier regulator of the γδT17 cell network provides a link between TCR signalling and the inhibition of the γδT17 cell fate.

Our results strongly support direct involvement of HEB in the gene network that drives γδT17 cell lineage choice, particularly in

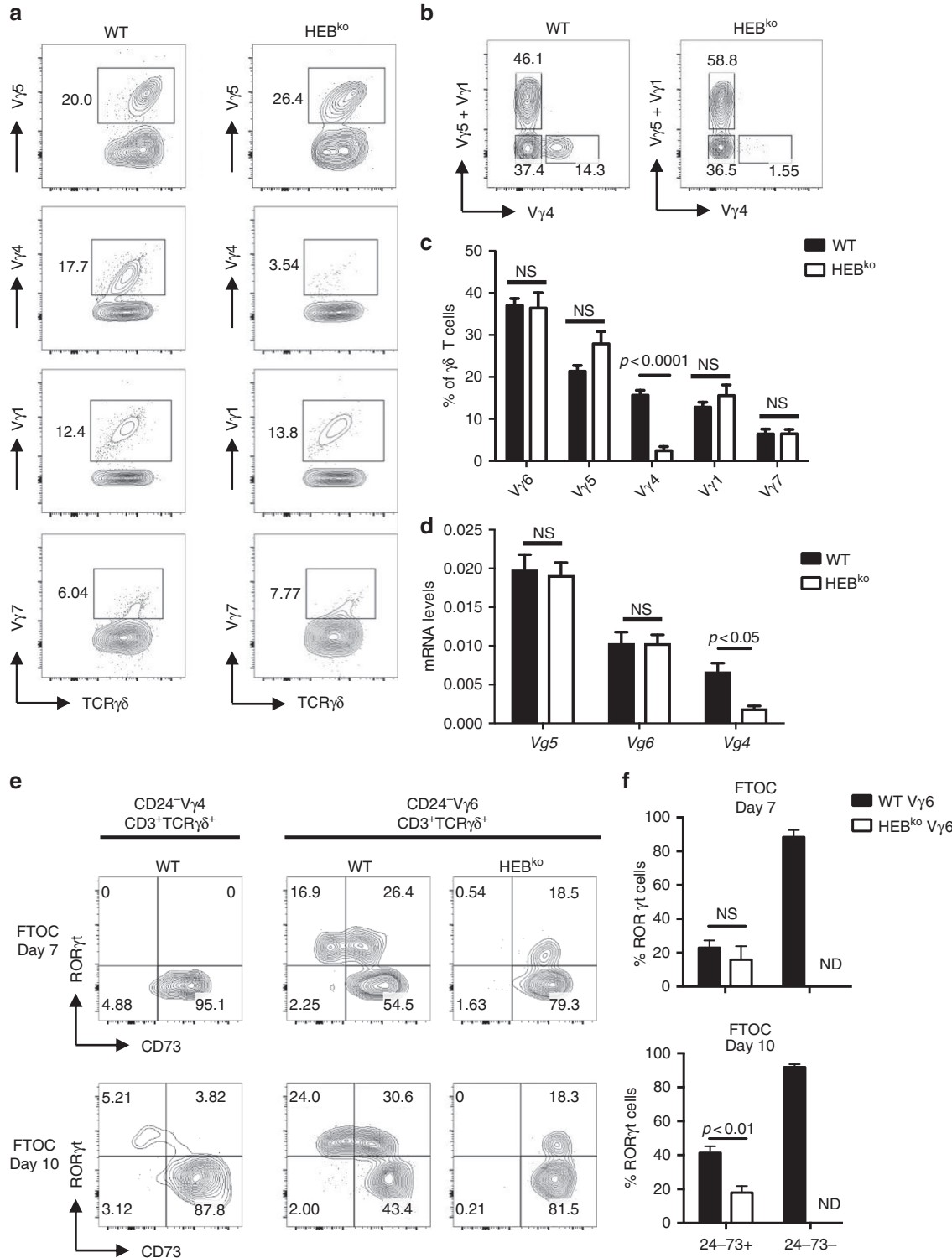

**Fig. 7** Multiple defects in Vγ4 and Vγ6 cells in HEB[ko] FTOCs. **a** Representative FACS plots showing the frequencies of Vγ subsets among all γδ T cells in WT and HEB[ko] d7 FTOCs. **b** Gating strategy for Vγ6 (Vγ5/1−Vγ4−CD3+TCRγδ+). **c** Quantification of frequencies of Vγ subsets among total γδ T cells in WT and HEB[ko] d7 FTOCs. **d** mRNA levels of Vγ transcripts in γδ T cells sorted from WT and HEB[ko] d7 FTOCs, as determined by qRT-PCR. Levels are relative to β-actin. **e** Intracellular expression of RORγt vs. surface CD73 in mature Vγ4 (CD3+TCRγδ+Vγ4+CD24−) from WT FTOCs, and mature Vγ6 (CD3+TCRγδ+Vγ5/4/1−CD24−) cells from WT and HEB[ko] FTOCs. **f** Quantification of the frequencies of RORγt+ cells among CD24−CD73+ Vγ6 cells and CD24−CD73−Vγ6 cells from WT and HEB[ko] FTOCs. All plots are gated on the CD3+TCRγδ+ population. Numbers in FACS plots indicate frequency within each gate. Data are representative of at least three independent experiments with at least 3 mice per group. Center values indicate mean, error bars denote s.e.m. p-values were determined by two tailed Student's t-test. ND = not done (due to low cell number). NS = not significant

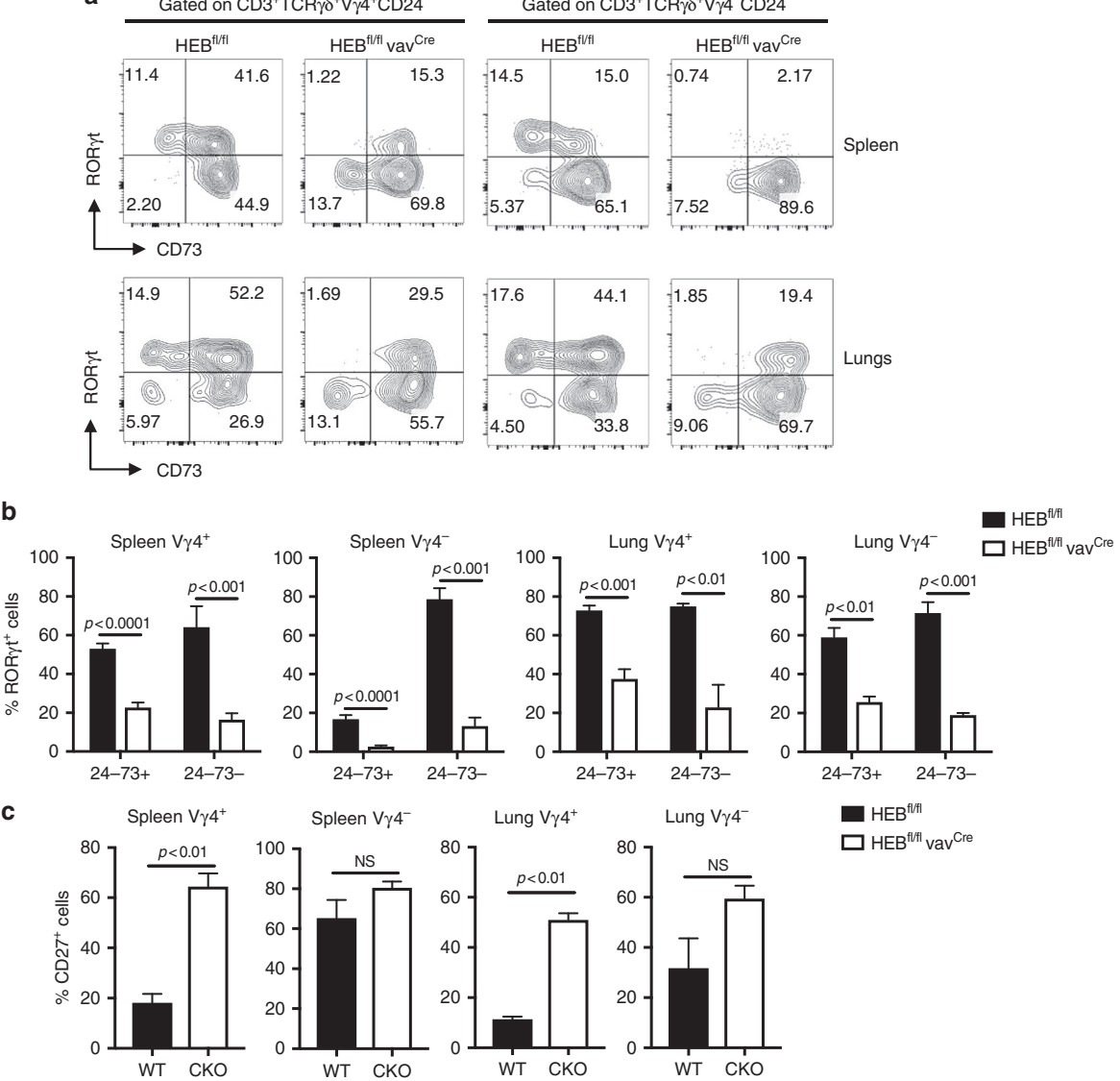

**Fig. 8** Defective RORγt expression in peripheral γδ T cells in adult HEB$^{cko}$ mice. **a** Representative FACS plots of intracellular RORγt vs. surface CD73 expression in mature Vγ4$^+$ and Vγ4$^-$ γδ T cells (CD24$^-$CD3$^+$TCRγδ$^+$) in the spleen and lungs of HEB$^{fl/fl}$ (WT) and HEB$^{fl/fl}$ vav$^{Cre}$ mice. Numbers in FACS plots indicate frequencies within each gate. **b** Quantification of the frequencies of RORγt$^+$ cells within CD24$^-$CD73$^+$ and CD24$^-$CD73$^-$ populations in Vγ4$^+$ and Vγ4$^-$ γδ T cells from the spleen and lungs of HEB$^{fl/fl}$ and HEB$^{fl/fl}$ vav$^{Cre}$ mice. **c** Quantification of the frequencies of CD27$^+$ cells within Vγ4$^+$ and Vγ4$^-$ γδ T cells in the spleen and lungs of HEB$^{fl/fl}$ and HEB$^{fl/fl}$ vav$^{Cre}$ mice, as determined by flow cytometry. All plots were gated on CD3$^+$TCRγδ$^+$ cells. Data are representative of at least two independent experiments with 3 mice per group. Center values indicate mean, error bars denote s.e.m. *p*-values were determined by two tailed Student's *t*-test. NS = not significant

the regulation of *Sox4* and *Sox13*, both of which have been linked to *Rorc* expression[15,16]. The similarities in phenotype between HEB-deficient mice and *Sox13*$^{-/-}$ mice further support a critical function of HEB upstream of *Sox13* expression[15,53]. In both HEB-deficient and *Sox13*$^{-/-}$ mice, Vγ4 γδT17 cells are absent in the adult thymus and periphery, and Vγ6 cells fail to behave as γδT17 cells in the fetal thymus. CD73 was not analyzed in these studies so whether *Sox13* deficiency affects Pathway 1, Pathway 2, or both, is unknown. HEB may also be indirectly involved in qualitative tuning of the γδ TCR signal, through *Sox13*-mediated expression of *Blk*, a tyrosine kinase specifically associated with γδT17 cell development[15,59]. The loss of HEB-mediated *Sox13* expression may result in impaired *Blk* expression and altered γδ TCR signalling outcomes.

After birth, and in late FTOC, RORγt$^+$ γδ T-cell precursors enter Pathway 1, in addition to Pathway 2. Subsequently, Pathway 2 is almost entirely lost in the adult thymus. This switch in developmental modes suggests changes in the types of γδ T-cell precursors, and/or in the thymic microenvironment. After the neonatal period, most of the RORγt$^+$ cells in the thymus are Vγ4$^+$ cells, which exhibit a varied TCR repertoire. These cells are likely exposed to an array of different ligands presented by the thymic epithelial cells, providing ample opportunity for positive selection through strong TCR signalling, which would be expected to bias cells toward the γδT1 cell fate. Therefore, the lack of Vγ4$^+$ γδT17 cells, coupled with the presence of CD27$^+$ Vγ4$^+$ γδT1s in the adult periphery, likely reflects a missed window of opportunity, rather than an ongoing defect in the adult thymus of HEB-deficient mice. The loss of access to Pathway 2 in the adult thymus could result from a lack of γδT17 cell-promoting signals[60,61], or from active inhibition by signals specific to the adult thymus. The latter hypothesis is consistent with a study in which Vγ6

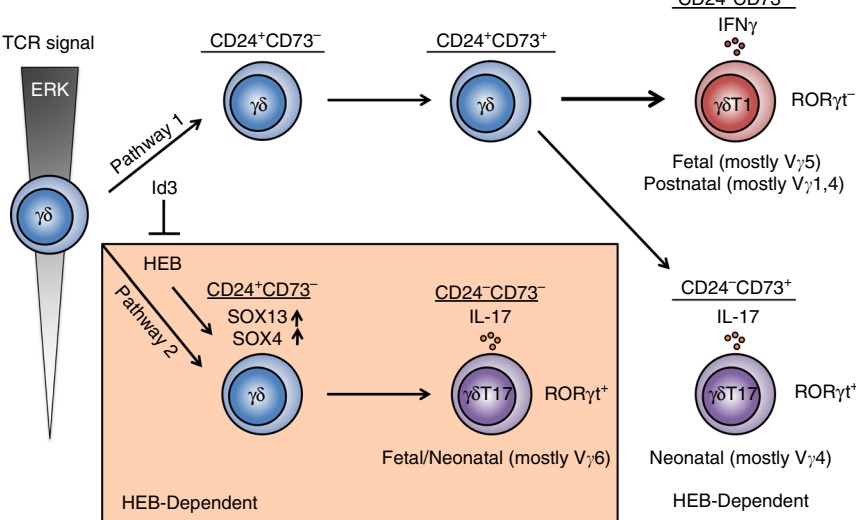

**Fig. 9** Model for two HEB-dependent pathways of γδT17 cell development. Uncommitted γδ T-cell precursors (circle on the left) encounter TCR signals that induce them to enter Pathway 1 (unboxed) or Pathway 2 (orange box), depending on the strength of ERK activity downstream of the TCR signal. Pathway 1 involves upregulation of CD73, and is favored by Id3 activity. Pathway 2 proceeds without CD73 upregulation, and entry into this pathway is dependent on HEB activity. High ERK signalling leads to the upregulation of Id3, which suppresses HEB function and thereby inhibits Pathway 2. Most cells in Pathway 1 become IFNγ-producing γδT1 cells. In the fetal thymus, these are mostly Vγ5+ cells, whereas they are mostly Vγ4+ and Vγ1+ cells in the postnatal thymus. A minority of Pathway 1 CD24+CD73+ cells become CD73+ γδT17 cells that are RORγt+ and express mostly Vγ4. HEB-deficient cells can take this pathway, but the resulting CD73+ RORγt+ cells are severely defective in IL-17 production. Pathway 1 generation of γδT17 cells is mostly restricted to the neonatal period. Pathway 2 leads to the generation of CD73− RORγt+ γδT17 cells that mostly express Vγ6, and this pathway is restricted to fetal and early neonatal life. Pathway 1 generation of γδT1 cells appears to be independent of HEB. Circles indicate cells at distinct developmental stages, arrows indicate developmental progression

+ γδT17 cell development was de-repressed in the adult thymus in a mouse model with altered corticomedullary thymic structure[62].

In summary, our studies show that HEB is critical for the initiation of the gene network that drives γδT17 cell development, and define multiple pathways by which this fate can be achieved. HEB is required at the earliest stages of γδ T-cell development for setting up the Sox-RORγt network that subsequently enables IL-17 production. We provide evidence that inhibition of HEB by Id3 leads to repression of the γδT17 cell network, linking it to γδ TCR-mediated repression of the γδT17 cell fate. In addition, our results suggest that HEB may function to alter aspects of the γδ TCR signal cascade, allowing the generation of CD24−CD73− γδT17 cells through Pathway 2. Therefore, our work reveals that HEB is central to a complex molecular network that controls γδT17 cell effector programming and maturation.

## Methods

**Mice**. WT, germline *Tcf12*−/− (HEB)[63], HEB[fl/fl][57], TCRδ-deficient[64], and vav[Cre][65] mice, all on the C57BL/6 background, were bred and maintained in the Comparative Research Facility of the Sunnybrook Research Institute (Toronto, Ontario, Canada) under specific pathogen-free conditions. The HEB-deficient strains, both germline (HEB[ko]) and conditional (HEB[cko]), lack endogenous expression of both HEBCan and HEBAlt, due to a deletion of a shared exon[41,57,63]. Timed mated pregnant CD1 mice were purchased from Charles River, Canada (Sherbrooke, Quebec, Canada). All animal procedures were approved by the Sunnybrook Research Institute Animal Care Committee.

**Tissue preparation**. Single-cell suspensions of spleen and/or thymus tissues of neonatal and adult mice 6–16-weeks old were prepared as previously described[21]. Lungs were minced and treated with collagenase IV (2 mg/ml) and DNase I (20 μg/ml) for 1 h at 37 °C, with rotation prior to disaggregation, and filtered through a 40 μm nylon cell strainers (Fisher Scientific, Missassauga, Ontario, Canada). Lung and spleen cells were treated with red cell lysis buffer prior to staining.

**Flow cytometry**. Antibodies were purchased from eBioscience (San Diego, California, USA), Biolegend (San Diego, California, USA) and BD Biosciences (Mississauga, Ontario, Canada). Vγ6-specific staining was accomplished in some experiments using undiluted supernatants from the 17D1 hybridoma (Sunnybrook Core Antibody Facility)[66]. A negative gating strategy was used in most experiments to identify Vγ6+ T cells, as previously described[67] and shown in Fig. 7b. Cells were washed and incubated with Fc blocking antibody (clone 2.4G2, Sunnybrook Core Antibody Facility, 1 mg/ml), followed by extracellular staining for surface CD4 (clone GK1.5), CD8α (clone 53–6.7), CD3 (clone 145-2C11), TCRγδ (clone GL3), TCRβ (clone H57–597), Vγ4 (Vγ2; clone UC3-10A6), Vγ5 (Vγ3; clone 536), Vγ1 (Vγ1.1; clone 2.11), Vγ7 (clone F2.67), CD27 (clone LG3-1A10), CD24 (clone M1/69) and/or CD73 (clone TY11.8). All flow cytometric analyzes were performed with a Becton-Dickenson (BD) LSRII, Diva software, and FlowJo software. Sorting was performed on a FACSARIA (BD). Representative gating strategies are shown in Supplementary Fig. 11.

**Intracellular cytokine and transcription factor staining**. Cells were stimulated by incubation for 5 h with IL-7 (10 ng/ml) alone; or IL-1β (10 ng/ml), IL-23 (10 ng/ml), IL-21 (20 ng/ml) and IL-7 (10 ng/ml); or PMA (50 ng/ml), Ionomycin (500 ng/ml) and IL-7 (10 ng/ml), in the presence of Brefeldin A (5 mg/ml; eBioscience). Cells were washed with 1XHBSS/BSA and incubated with Fc block before staining for surface CD3, TCRγδ, Vγ4, Vγ5, Vγ1, CD27, CD73, and/or CD24 (see flow cytometry section for clones). Cells were fixed and permeabilized (Fix and Perm Cell Permeabilization Kit; eBioscience) and stained with antibodies against IL-17A (clone eBio17B7) or IFNγ (clone XMG1.2). For intranuclear staining for RORγt (clone B2D), cells were fixed and permeabilized (FoxP3 Staining Kit, eBioscience).

**ELISA**. FACS-sorted γδ T cells from E14.5 FTOC day 7 were stimulated with IL-7 (10 ng/ml) alone; or IL-1β (10 ng/ml), IL-23 (10 ng/ml), IL-21 (20 ng/ml) and IL-7 (10 ng/ml), for 72 h. Supernatants were collected, and IL-17A protein levels were evaluated by ELISA, using DuoSet ELISA kits (catalog # DY421-05, R&D Systems, Burlington, Ontario, Canada), as per the manufacturer's instructions.

**Retroviral transduction**. Cells from d5 WT FTOCs were co-cultured overnight with stable retrovirus-producing GP + E.86 packaging cells, as previously described[68]. Following this step, the transduced (GFP+) CD45+TCRγδ+CD3+CD24+CD73− cells were purified by cell sorting, and RNA was extracted for gene expression analysis.

**Fetal thymus organ culture**. Mice were set-up for timed matings to obtain WT and HEB[ko] or HEB[cko] littermates. Fetal thymic lobes were isolated at E14.5 and cultured under standard FTOC conditions using gelfoam sponges and Millipore filters as previously described[69]. After 4–10 days, the lobes were treated with

collagenase for 30 min and subjected to mechanical disaggregation to make single-cell suspensions. For the precursor-product experiments (Fig. 2), TCRδ-deficient fetal lobes from E14.5 embryos were treated with 1.35 mM deoxyguanosine for 2 days and then rested in media before being placed in hanging drop culture with γδ T-cell developmental subsets sorted from traditional FTOCs.

**Reverse transcription and real-time PCR.** Total RNA was isolated from FACS-sorted cell populations using Trizol (Invitrogen, Burlington, Ontario, Canada) and converted into cDNA using Superscript III (Invitrogen). Quantitative real-time PCR reactions were performed with SYBR green (Bio-Rad, Mississauga, Ontario, Canada) and 2.5 µM of gene-specific primers. Reactions were run and analyzed using an Applied Biosystems (Foster City, California, USA) 7000 sequence detection system. All PCR reactions were done using the same serially diluted cDNA samples, and values were normalized to a β-actin-specific signal. Relative values were calculated using the delta Ct method[70]. Primer sequences are shown in Supplementary Table 1.

**Luciferase reporter assay.** A *Sox13*-reporter construct was generated which contained a region upstream of the second exon of the *Sox13* locus by PCR amplification from genomic DNA, and cloning into the *KpnI* and *XhoI* sites upstream of the luciferase coding region of pGL4.10 (Promega, Madison, Wisconsin, USA), using Gibson Assembly (New England Biolabs, Ipswich, Massachusetts, USA). Gibson primers used for PCR were designed to overlap with pGL4.10 ends. Primer sequences are given in Supplementary Table 1. The construct confirmed by sequencing. HEK293T cells were co-transfected with the reporter construct, *Renilla* control vector (pRL-null), and pCMV-based expression constructs encoding HEBAlt or HEBCan, or an empty vector, using Lipofectamine 3000 (Fisher Scientific). Firefly and *Renilla* luciferase were detected using the Dual Luciferase assay kit, according to manufacturer's instructions (Promega). Data were normalized to *Renilla* luciferase activity to account for transfection efficiency.

**Chromatin immunoprecipitation.** Rag-2-deficient DN3 cells were derived from fetal liver progenitors on OP9-DL4 monolayers as previously described[32]. Cells were stained using fluorochrome-coupled antibodies, fixed for 10 min at 20 °C with 1% (wv/vol) formaldehyde, sorted on a FACSAria II (BD Biosciences), lysed, and sonicated. Sonicated chromatin was immunoprecipitated with 10 µg anti-HEB (affinity-purified rabbit polyclonal sera raised against the last 12 amino acids of the C-terminus of HEB). Samples were washed, and bound chromatin was eluted. Crosslinking was reversed overnight at 65 °C. Samples were treated with RNase A and proteinase K, and DNA was purified using the ChIP DNA Clean and Concentrator Kit (Zymo, Irvine, California, USA). Quantitative real time PCR was performed with QuantiTect SYBR Green PCR Kit (Qiagen, Toronto, Ontario, Canada) on an ABI Prism 7700 Real-Time PCR machine (Applied Biosystems). Primer sequences are given in Supplementary Table 1.

**Statistical analysis.** Data were compared by two tailed Student's *t*-test, or by one-way ANOVA (Fig. 1), with $p < 0.05$ having statistical significance. Error bars represent the estimate of variation as the standard error of the mean (SEM). "NS" indicates a non-statistically significant difference. All experiments represent at least two independent experiments with at least three biological replicates each. Variance was similar between groups. Due to the uncertain nature of timed matings of heterozygotes in obtaining the correct genotypes at the correct embryonic age, data were obtained over time and pooled into like groups for analysis, with each thymic lobe or mouse representing a biological replicate. No randomization was used, and blinding was not done.

**Data availability.** The data that support the findings of this study are available from the corresponding author upon request.

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

## Acknowledgements

We would like to thank Dr. Yuan Zhuang and Dr. Trang Hoang for generously providing HEB^fl/fl and HEB^ko mice, and Drs. Rebecca O'Brien and Robert Tiggelaar for the 17D1 antibody hybridoma. We would also like to thank the Centre for Flow Cytometry and Microscopy and Comparative Research at the Sunnybrook Research Institute (SRI) for technical support. This work was funded by the Canadian Institutes for Health Research (MOP 82861 and PJT 153058) to M.K.A., and NIH-1P01AI102853-01 to D.L.W. and J.C. Z.-P. J.C.Z.-P. was supported by a Canada Research Chair in Developmental Immunology. T.S.H.I. was funded by an Ontario Graduate Scholarship.

## Author contributions

T.S.H.I. designed and performed experiments, analyzed data and wrote the manuscript. S.F. performed the ChIP experiments. A.J.M. established the HEB^fl/fl Vav^cre mouse strain. A.T.-G. cloned the reporter constructs and performed the luciferase assays. P.Z. assisted in experimental design and assay optimization, and provided Id3 retroviral packaging cell lines. E.L.Y.C. assisted in experiments. D.W. and J.C.Z.-P. provided input on experimental design and interpretation of data. M.K.A. designed experiments, analyzed data, and wrote the manuscript. All authors provided editorial input on the manuscript.

## Additional information

**Competing interests:** The authors declare no competing financial interests.

