## [Peer Review File · Nature Communications]

Reviewers' comments:

Reviewer #1 (Remarks to the Author):

In this paper, In et al. show that the transcription factor HEB selectively participates in the development of IL-17-producing gd (gd17) T cells, for both Vg4+ and Vg6+ subsets. The role of HEB in the development of gdT17 cells is interesting and novel, but unfortunately the manuscript lacks substantial information on the underlying mechanism(s). Thus, although solid in terms of cellular immunology and phenotype characterization, the paper fails to establish the key molecular links between HEB and the remaining transcriptional network that controls gdT17 cell development.

Major concerns:

- Although the present work somehow brings HEB into the pathway that controls the generation of gdT17 cells, the results, as well as the discussion, don't provide a convincing integration of the available data on "how TCR signaling impacts the Sox-Tcf network, within the unique context of developing $\gamma\delta$ T cells"(page 4 Line 66). The role of TCR signalling should be formally tested in the current study, either by providing agonist antibodies in FTOC or using in vivo models that interfere with TCR signal strength (as those developed by co-author David Wiest). These data would be necessary to support the following conclusions on P13 Line 232: "Furthermore, our work provides a novel link between TCR signaling and the $\gamma\delta$ T17 network, by revealing the intersection of TCR-mediated Id3 upregulation and HEB-dependent activation of the $\gamma\delta$ T17 transcriptional network containing Sox13, Sox4, and ROR γ t." (...) "Our data support a model in which HEB directs the Sox13-Sox4-ROR γ t cascade that is necessary for $\gamma\delta$ T17 programming in immature $\gamma\delta$ TCR-bearing cells, which proceeds in the absence of a strong TCR signal (Fig. 7)." Furthermore, this interpretation should be qualified in light of the recent Nat Immunol paper (Munoz-Ruiz et al. 2016, cited as ref. 13) showing distinct TCR signaling requirements of Vg4+ and Vg6+ thymocytes.

- Although the authors state (Page 14 Line 261) that "Current studies are focused on determining whether HEB factors directly regulate Sox13 or Sox4, and whether it collaborates with SOX factors to directly regulate ROR γ t in $\gamma\delta$ T-cells", the question of how HEB regulates Sox 13 transcription is well within the scope of the current study since they are necessary to support the conclusions drawn by the authors. Possible approaches would be HEB overexpression (Takeuchi et al); creation of reporter cell lines to assess HEB transcriptional activity on the promoter sequences of Sox13, Sox4 and Rorc; ChIP-qPCR or ChIP-seq data using anti-HEB to pull down chromatin from gd T cells and determine binding to the sequences of Sox13, Sox4 and Rorc (Yoon et al). This mechanistic insight is critical because HEB can either promote Sox13 transcription (as proposed) or, alternatively, be required for survival/ selection of Sox13+ thymocytes.

References:

- Takeuchi et al, "E2A and HEB Activate the Pre-TCR Promoter During Immature T Cell Development" *The Journal of Immunology*, 2001, 167: 2157–2163.
- Yoon et al. HEB and E2A function as SMAD/FOXH1 cofactors. *Genes Dev.* 2011 Aug 1; 25(15): 1654–1661.

- Another major question that remains unresolved: what induces HEB expression?

- Page 14 Line 258: "Combined, these results indicate that HEB and Sox13 may act primarily during development rather than during peripheral activation of $\gamma\delta$ T17s, where ROR γ t may be sustained without them." This conclusion should be qualified since the Authors did not formally address HEB function in peripheral gdT17 cells, for example by HEB^{fl/fl} mice with IL17-Cre mice, and checking gd17T cell maintenance and activation upon in vivo challenge. While this experiment may well be beyond the scope of the current study, the Authors could easily document HEB expression in peripheral versus thymic gd T cell subsets (segregated on the basis of CD44, CD27 and CCR6 expression).

- The main approach used in this study was to perform FTOC for 7 days with plain Heb^{-/-} mice or to analyse the adult thymus and spleen from HEB^{f1/fl} x vav^{Cre} mice. In Zhuang, Y., et al. homozygous mice for the HEB mutation were reported to die within the first 2 weeks after birth. It is therefore unclear why direct (ex vivo) analysis of foetal thymus was not performed, for the pick of gdT17 cells appears around E16-18 until neonates with about 40% of the total gd T cells that produce IL-17 (see Haas, et al.). These ex vivo data could be highly relevant.

References:

- Zhuang, Y., et al. B-lymphocyte development is regulated by the combined dosage of three basic helix-loop-helix genes, E2A, E2-2, and HEB. *Mol Cell Biol* 16, 2898-2905 (1996).
- Haas, J.D. et al. Development of interleukin-17-producing gammadelta T cells is restricted to a functional embryonic wave. *Immunity* 37, 48-59 (2012).

Minor Points

- Figure 1C: the standard deviations are missing on the two graphs.
- Figure 1E & F: The statistical p values (or not significant "ns") are missing for Rorc, Egr3, Tbx21, Ifng.
- Page 11 line 201: "Strikingly, lower proportions of V γ 4+ cells in the adult HEB^{-/-} thymus expressed ROR γ t and the $\gamma\delta$ T17 marker CCR6 than in WT, whereas the percentages of the V γ 4+ cells expressing CD27 were increased (Fig. 5e, f)." As the figure 5f does not show increase in CD27+V γ 4+ T cells, the text should be corrected to: "Strikingly, ..., whereas the percentages of the V γ 4+ cells expressing CD27 were unaltered (Fig. 5e, f)..."

Reviewer #2 (Remarks to the Author):

This study demonstrates the importance of the E protein transcriptional regulator HEB to direct development of a subset of $\gamma\delta$ T cells that is a significant source of IL-17 at barrier surfaces ($\gamma\delta$ T17). The data are generally of good quality and the observations are interesting. But the study is framed as distinguishing two models of $\gamma\delta$ T17 lineage fate determination, but it does not, as linkage of TCR signaling and Id3 induction in relation to HEB activity is not addressed here. Moreover, the study seems somewhat preliminary, as it is lacking a strong mechanistic understanding for many of the observations, and does not characterize a potential novel developmental pathway in generation of 24-73- cells in any detail.

Some specific issues:

1. All data should include statistics (for example, missing from parts of Fig. 1, Supp. Fig. 1, Fig 3). In this regard it is stated that Egr3 and Ifng are not reduced in HEB-deficient cells. Are the obvious changes in mean expression in Fig. 1f not significant? Is the data simply limited by the n?
2. Surprisingly, given the hypothesis, expression of Id3 is not assessed but should be throughout (Fig. 1,3).
3. The example in Fig. 2a, does not appear to be representative of the data in Fig. 2b in regard to P/I induced IFN γ producers. Also, data in Fig. 3a does not appear to be representative of the compiled data in Fig. 3b (24+73-). Same for Fig. 5 e,f.
4. The authors argue that loss of Sox13 and Sox4 in 24+73- cells is 'indicative of a role for HEB in $\gamma\delta$ T17 programming prior to TCR signaling". Later stages show loss of HEB and Sox13/Sox4, including in 24-73- cells that express the highest Rorc, the proposed downstream target of the Sox factors. How this fits the model is not addressed. (Are SOX factors proposed as pioneer factors? What is the evidence?). Is there indeed any loss of RORc in HEB-deficient 24+73- cells (no stats are given)? Could the 24+73- cell population be heterogeneous, with a mix of precursors? Moreover, given the loss of HEB itself through these developmental stages, how does Id3 fit into

the picture?

5. The precursor product relationship in production of the 24-73- subset is not directly addressed. Can they be generated from 24-73+ and 24+73-cells?

6. Some of the V γ 6 data is indirect (Fig. 4) and some direct (Fig. 6). Are there technical reasons for this?

7. The loss of ROR γ t cells in Fig. 5f is less than 2-fold. What is the efficiency of deletion of HEB in this system? What are cell numbers?

8. Gut as well as lung cells should be shown, as changes in migratory properties or responses to microenvironmental cues of HEB-deficient cells is possible.

9. The pdf was difficult to read, but it appears that there is no loss of ROR γ t+V γ 4- cells in Fig. 6b. What is the explanation, particularly given Fig. 6c? Also, the effect on lung V γ 4- cells is less than 2-fold. Is this indicative of poor deletion or only a modest effect of HEB deficiency?

Point-by-point Response to Reviewers:

We thank the reviewers and the editors for giving us the chance to revise and resubmit our manuscript, "Requirement for HEB in the specification of fetal IL-17-producing $\gamma\delta$ T cells", by In *et al.* As part of the revisions, we have performed many additional experiments and have made extensive modifications to the manuscript, adding more detailed analysis of ROR γ t (*Rorc*) expression as a marker of $\gamma\delta$ T17 programming, and showing analyses at day 10 as well as day 7 FTOCs. We have reconsidered the hypothesis that it is TCR-mediated up-regulation of Id3 that inhibits the $\gamma\delta$ T17 fate, and have instead focused on characterizing the novel CD24-CD73- $\gamma\delta$ T17 population that we have discovered, and how HEB integrates into the gene network that drives $\gamma\delta$ T17 programming. We feel that this refocus strengthens the paper considerably, and provides mechanistic evidence that HEB is a direct player in this network.

New and revised sections are marked in yellow in the manuscript. Many of the original figures have been moved to Supplementary Information to make room for the new experiments.

Reviewer #1 (Remarks to the Author):

The role of TCR signalling should be formally tested in the current study, either by providing agonist antibodies in FTOC or using in vivo models that interference with TCR signal strength (as those developed by co-author David Wiest).

- We have now chosen to refocus and expand our studies on the role of HEB in $\gamma\delta$ T17 lineage choice, and rather proposing that HEB is a mediator of a $\gamma\delta$ T17/ $\gamma\delta$ T1 choice involving differential TCR signaling. Further experiments using a fixed TCR on a Rag2-/- HEB-/- background will be needed to directly examine the interplay between the TCR, Id3, and HEB.

Furthermore, this interpretation should be qualified in light of the recent Nat Immunol paper (Munoz-Ruiz et al. 2016, cited as ref. 13) showing distinct TCR signaling requirements of Vg4+ and Vg6+ thymocytes.

- We agree and now relate our results to the observations of Munoz-Ruiz *et al.* in the discussion section. However, we are no longer invoking TCR signaling as an upstream event in repression of the $\gamma\delta$ T17 lineage in the context of the fetal thymus, but focus instead the balance between Id3 and HEB in CD24+ $\gamma\delta$ T cell precursors as the prime determinant of Sox13 and Sox4 expression (**Fig. 4C**).

The question of how HEB regulates Sox 13 transcription is well within the scope of the current study since they are necessary to support the conclusions drawn by the authors. Possible approaches would be HEB overexpression (Takeuchi et al); creation of reporter cell lines to assess HEB transcriptional activity on the promoter sequences of Sox13, Sox4 and Rorc; ChIP-qPCR or ChIP-seq data using anti-HEB to pull down chromatin from gd T cells and determine binding to the sequences of Sox13, Sox4 and Rorc (Yoon et al). This mechanistic insight is critical because HEB can either promote Sox13 transcription (as proposed) or, alternatively, be required for survival/ selection of Sox13+

thymocytes.

- We agree that this is an important mechanistic component of our study, and have performed modified versions of all of these approaches, with considerable success. We now show that HEB occupies a site in the *Sox4* locus in Rag2^{-/-} DN3 cells (**Fig. 5A, B**), and that a region in the *Sox13* locus upstream of the first exon can be activated by HEB factors using a reporter assay (**Fig. 5C, D**). HEBCan overexpression was not successful, in that overall levels of HEBCan were not increased even in GFP⁺ cells (not shown), but our Id3 overexpression studies (**Fig. 4C**) are consistent with direct activation of the Sox genes by HEB factors.

Another major question that remains unresolved: what induces HEB expression?

- We are very interested in this question, and have shown that HEBAlt is up-regulated by HEBCan and in response DLL-Notch signaling in DN thymocytes. However, for the purposes of this study, we propose it is the balance between Id3 and HEB factors that controls fate within the CD24⁺ cells, rather than absolute HEB expression levels.

“Combined, these results indicate that HEB and Sox13 may act primarily during development rather than during peripheral activation of $\gamma\delta$ T17s, where ROR γ t may be sustained without them.” This conclusion should be qualified. The Authors could easily document HEB expression in peripheral versus thymic gd T cell subsets (segregated on the basis of CD44, CD27 and CCR6 expression).

- We thank the reviewer for this suggestion. We have performed these expression studies and have found that HEB levels are present at low levels in peripheral $\gamma\delta$ T cells subsetted by V γ chains similar to their expression in mature CD73⁻ $\gamma\delta$ T cells in fetal thymus (not shown). Therefore, we have deleted this statement pending further experiments.

The main approach used in this study was to perform FTOC for 7 days with plain Heb^{-/-} mice or to analyse the adult thymus and spleen from HEB^{fl/fl} x vavCre mice. In Zhuang, Y., et al. homozygous mice for the HEB mutation were reported to die within the first 2 weeks after birth. It is therefore unclear why direct (ex vivo) analysis of foetal thymus was not performed, for the pick of gdT17 cells appears around E16-18 until neonates with about 40% of the total gd T cells that produce IL-17 (see Haas, et al.). These ex vivo data could be highly relevant.

- Our HEB^{-/-} mice die before birth (now clarified in the text), which is due to backcrossing of the original 129/B6 strain onto the B6 background, but we were able to recover several embryos at E17.5. The lack of HEB^{-/-} Vg4⁺ cells in E17.5 ex vivo fetal thymus is shown in **Fig. S7A**, we show in **Fig. S3A,B** that E17.5 WT and HEB KO ex vivo thymocytes have the same phenotypes that we present for FTOC data in **Fig. 3**.

Minor Points

Figure 1C: the standard deviations are missing on the two graphs.

- The error bars are present but too small to be easily seen.

Figure 1E & F: The statistical *p* values (or not significant “ns”) are missing for *Rorc*, *Egr3*, *Tbx21*, *lfng*.

- *P* values are now present for all significant differences, and “n.s.” depicted on the non-significant differences for all relevant figures.

Page 11 line 201: “Strikingly, lower proportions of $V\gamma 4^+$ cells in the adult *HEB*^{-/-} thymus expressed *ROR γ t* and the $\gamma\delta T17$ marker *CCR6* than in WT, whereas the percentages of the $V\gamma 4^+$ cells expressing *CD27* were increased (Fig. 5e, f).” As the figure 5f does not show increase in *CD27*⁺ $V\gamma 4^+$ T cells, the text should be corrected to: “Strikingly, ..., whereas the percentages of the $V\gamma 4^+$ cells expressing *CD27* were unaltered (Fig. 5e, f)...”

- This has been fixed in what is now **Fig. 7**.

Reviewer #2 (Remarks to the Author):

*This study demonstrates the importance of the E protein transcriptional regulator HEB to direct development of a subset of $\gamma\delta T$ cells that is a significant source of IL-17 at barrier surfaces ($\gamma\delta T17$). The data are generally of good quality and the observations are interesting. But the study is framed as distinguishing two models of $\gamma\delta T17$ lineage fate determination, but it does not, as linkage of TCR signaling and *Id3* induction in relation to HEB activity is not addressed here. Moreover, the study seems somewhat preliminary, as it is lacking a strong mechanistic understanding for many of the observations, and does not characterize a potential novel developmental pathway in generation of 24-73-cells in any detail.*

- Given additional data and new insights, we have now chosen to focus on the contribution of HEB factors to the programming of $\gamma\delta T17$ cells, and not on a $\gamma\delta T1/\gamma\delta T17$ lineage choice. We have also defined the developmental pathway that leads to the generation of *CD24*-*CD73*- $\gamma\delta T17$ s.

Some specific issues:

1. All data should include statistics (for example, missing from parts of Fig. 1, Supp. Fig. 1, Fig 3). In this regard it is stated that *Egr3* and *lfng* are not reduced in *HEB*-deficient cells. Are the obvious changes in mean expression in Fig. 1f not significant? Is the data simply limited by the *n*?

- Statistics are now included, both *p* values for statistically significant differences and “n.s.” for non-significant differences. Additional experiments have been conducted to raise the sample number and increase the accuracy, shown now in **Fig. 4**. In short, *Egr3* is not different, and *Tbx21* is higher in one of the *HEB*^{-/-} subsets but otherwise unchanged.

2. Surprisingly, given the hypothesis, expression of *Id3* is not assessed but should be throughout (Fig. 1,3).

- We have now included Id3 expression analysis in the characterization of WT and HEB KO CD24/CD73 subsets (**Fig. 1, 4**).

3. The example in Fig. 2a, does not appear to be representative of the data in Fig. 2b in regard to P/I induced IFN γ producers. Also, data in Fig. 3a does not appear to be representative of the compiled data in Fig. 3b (24+73-). Same for Fig. 5 e,f.

- Both single producers and double producers are included in the calculations used to make the graphs. This is now indicated in the relevant figure legends, for clarity.

4. The authors argue that loss of Sox13 and Sox4 in 24+73- cells is ‘indicative of a role for HEB in $\gamma\delta$ T17 programming prior to TCR signaling’. Later stages show loss of HEB and Sox13/Sox4, including in 24-73- cells that express the highest Rorc, the proposed downstream target of the Sox factors. How this fits the model is not addressed. Are SOX factors proposed as pioneer factors? What is the evidence?

- It has been recently reported that TCF1 contains endogenous HDAC activity, which represses gene expression. By extension, if TCF1 activity represses ROR γ t, then SOX up-regulation could inhibit this activity, leading to derepression of ROR γ t. This notion is now part of the discussion.

Is there indeed any loss of ROR γ c in HEB-deficient 24+73- cells (no stats are given)? Could the 24+73- cell population be heterogeneous, with a mix of precursors?

- After many more experiments, we have still found no statistical difference in Rorc mRNA expression in WT vs HEB-/- CD24+CD73- in FTOC day 7. The CD24+CD73- population is likely very heterogeneous (certainly in terms of V γ chain expression, shown in **Fig. S5**), and the timing of embryo age (hours before or after “E14.5”) could influence these numbers.

Moreover, given the loss of HEB itself through these developmental stages, how does Id3 fit into the picture?

- Id3 also decreases during development (now shown in **Fig. 1E** and **Fig. 4**), so we believe the most relevant comparison between Id3 and HEB factors is at the CD24+CD73- and CD24+CD73+ stages, where lineage choice is most likely occurring.

5. The precursor product relationship in production of the 24-73- subset is not directly addressed. Can they be generated from 24-73+ and 24+73-cells?

- We thank the review for this excellent suggestion. We have now done this experiment (**Fig. 2**), and show that only CD24+CD73- cells give rise to CD24-CD73- cells.

6. Some of the V γ 6 data is indirect (Fig. 4) and some direct (Fig. 6). Are there technical reasons for this?

- Yes, this is correct, the 17D1 antibody staining is quite poor (see **Fig. S8**). The indirect method has been validated in the past (Sheridan, 2013), and that is the method we used for all figures except Figure S8.

7. *The loss of ROR γ t cells in Fig. 5f is less than 2-fold. What is the efficiency of deletion of HEB in this system?*

- The severe effect on $\alpha\beta$ T cells in the HEB CKO, which is driven by Vav-Cre, indicates that deletion is quite efficient (not shown). Subsetting the cells into CD73+ and CD73- and intracellular staining of ROR γ t helped to clarify the differences in protein expression; there is no statistically significant difference in the CD73+ subset at FTOC day 7, but there is a clear decrease in HEB-/- CD73+ cells at day 10. Importantly, we optimized the ROR γ t intracellular staining and now it is much stronger, providing clearly positive and negative populations in **Fig. 3**, **Fig. 6**, and **Fig. 7**.

8. *Gut as well as lung cells should be shown, as changes in migratory properties or responses to microenvironmental cues of HEB-deficient cells is possible.*

- We agree that this is a very interesting question but feel it is beyond the current scope of the paper.

9. *The pdf was difficult to read, but it appears that there is no loss of ROR γ t+V γ 4- cells in Fig. 6b. What is the explanation, particularly given Fig. 6c? Also, the effect on lung V γ 4- cells is less than 2-fold. Is this indicative of poor deletion or only a modest effect of HEB deficiency?*

- Apologies for the poor quality of the PDF. We have revised this figure, now as **Fig. 7**, with graphs that clearly show the differences in ROR γ t+ cells in CD73+ and CD73- cells in the periphery. In the future, we hope to do these experiments in the context of a TCR $\gamma\delta$ -specific HEB deletion, to alleviate any homeostatic expansion of $\gamma\delta$ T cells that might occur in HEB-deficient mice.

Reviewers' comments:

Reviewer #1 (Remarks to the Author):

The revised paper by In et al. contains some new interesting data (such as Fig 5, which nicely addresses one of my previous main concerns); and attempts to refine the proposed model (while departing from a TCR signaling model). In that regard, the focus on the fetal CD24- CD73- gd subset is interesting but not particularly convincing as it is. It would be critical to show the relevance of ex vivo CD24- CD73- gd thymocytes (within the fetal gdT17 compartment), otherwise this population comes across as mostly an FTOC "artifact": note that it constitutes >30% at day 10 FTOC versus only 8% in E17.5... To show relevance, authors can gate on IL-17 producing (or RORgt+) ex vivo fetal thymocytes and show CD24 vs CD73 plots. I think Fig 1 needs to be improved to make it a strong(er) start to the paper.

At the end of the paper, there is another aspect that needs additional clarification, which has to do with how "HEB is critical for gdT17 generation through multiple developmental pathways" (lines 332-3). As the authors discuss (lines 327-332), HEB seems to control: the development of fetal CD24- CD73- RORgt+ Vg6+ thymocytes; a later fetal CD73+ RORgt+ thymocyte subset; fetal Vg4+ and adult RORgt+ Vg4+ thymocytes. What unifies these distinct pathways in their HEB dependence? On the other hand, what are the HEB-independent gdT17 pathways and (how) do they rely on RORgt and Sox4/13 function? I think the authors should further clarify their proposed model on gdT17 development for the benefit of the reader.

Minor issues:

- The Vgamma usage (Vg1/4/6) of the new fetal population (at E17.5 ex vivo) would be clearly shown in Fig 1.
- Although Il17a PCR data are shown (Fig 1d), i.e. IL-17A (versus IFN-gamma) and RORgt protein staining in E17.5 ex vivo CD24- CD73- gd thymocytes should be added in order to define them as bonafide gdT17 cells ex vivo.
- Have the authors analyzed CD24- CD73- gd thymocytes in newborn or young mice? How does this subset behave throughout ontogeny (ex vivo)? Only E17.5 and 12 w.o. thymi are shown, thus leaving a huge gap in between...
- Each paragraph describing the results should end with a clear conclusion, as to help the reader to follow the authors as they build their developmental model.

Reviewer #2 (Remarks to the Author):

This remains a well performed study that demonstrates the complex role of HEB in $\gamma\delta$ T cell development in the thymus. In this revision, the authors have turned away from the original thrust of the paper, which was not well addressed by the data presented, that HEB is a key downstream mediator that integrates TCR signaling to regulate cell fate decisions. Thus, although the study very convincingly shows that HEB is critical for aspects of $\gamma\delta$ T17 development, and demonstrates fetal and adult differences, there is somewhat reduced impact compared to the original goal of this study. In this regard, the biggest limitation is not knowing how the balance of E proteins and inhibitors leads to functional outcome as proposed. It was brought up in the previous review that the cells that express the lowest Sox4, Sox13, Id3, and HEB express the highest Rorc and Il17a. The authors now propose some role for Tcf7 to explain this. But as the pattern of expression or activity of Tcf7 is not addressed, it is difficult to know how this resolves this apparent paradox. Moreover, if E protein activity is upstream of Sox4 and Sox13, then E proteins are active in the

cells with the highest Id3 (as well as HEB), again leading to the difficulty in inferring overall E protein activity without some actual measure. It is certainly a reasonable hypothesis that HEB-dependent upregulation of Sox4 and Sox13, along with downstream Tcf7, leads to repression of CD24 and increased Rorc expression coincident with the downregulation of the SOX factors in a subset of cells (what controls formation of this subset is unclear). It is just that the data do not reveal this, nor how this complex network works. (Single cell RNA-seq and epigenomic studies might be required to unravel this.)

Overall, the authors have shored up the original findings and provided some new data, most notably that SOX4 and possibly SOX13 are likely direct targets for regulation by HEB/ E proteins and, consistent with that, that ectopic expression of Id3 can repress SOX4 and SOX13 expression. An additional key finding is that the CD24⁺CD73⁻ cell population contains precursors for CD24⁻CD73⁻ $\gamma\delta$ T17 cells. These are significant results based on technically challenging experiments, and greatly enhance the paper.

The fetal (complete HEB KO in FTOC) and adult (cHEB KO) analyses show differences in V γ 4 that likely reflect the complexity of the biology. This is interesting, although mechanistically not explored.

Minor notes:

Failure to look in the gut in cKO mice remains a weakness.

The reviewer assumes that the authors are talking about the TCF-1 protein, which is encoded by Tcf7, not Tcf1 (line 343).

Error bars are missing from Fig. 1d, II17a.

Periods should replace commas in numbers on Figures (?)

Page 10, reference should be to Supp. Fig. 4 (not 3d)

Fig. 4b, Id3⁻ is n.s. correct for first 2 cell populations?

Response to Reviewers

Re: NCOMMS-16-21340B

Please note: All major revisions are highlighted in yellow in the manuscript. In particular, Figure 1 has been extensively revised, and Figure 3 and Supplementary Figure 1 are new. The majority of the discussion has been rewritten to reflect the new data and its impact on our understanding of $\gamma\delta$ T17 development.

Reviewer #1 (Remarks to the Author):

The focus on the fetal CD24⁻ CD73⁻ gd subset is interesting but not particularly convincing as it is. It would be critical to show the relevance of ex vivo CD24⁻ CD73⁻ gd thymocytes (within the fetal gdT17 compartment), otherwise this population comes across as mostly an FTOC “artifact”: note that it constitutes >30% at day 10 FTOC versus only 8% in E17.5. To show relevance, authors can gate on IL-17 producing (or ROR γ t⁺) ex vivo fetal thymocytes and show CD24 vs CD73 plots. I think Fig 1 needs to be improved to make it a strong(er) start to the paper.

We thank the reviewer for these suggestions. We have spent considerable time examining the $\gamma\delta$ T-cell subsets in *ex vivo* thymocytes from fetal (E17.5) and neonatal (day 1 and 7) mice, and evaluated their distribution in the ROR γ t⁺ and ROR γ t⁻ fractions (Fig 1). We also show the CD24/CD73 subsets gated on IL-17 producers in Supplementary Fig. 1, which supports the ROR γ t data. Finally, we found that the WT lung and spleen contain CD24⁻CD73⁻ $\gamma\delta$ T-cells, whereas the gut does not. We feel that this new data leaves no doubt that the CD24⁻CD73⁻ $\gamma\delta$ T-cell subset is a physiologically relevant population.

The Vgamma usage (Vg1/4/6) of the new fetal population (at E17.5 ex vivo) would be clearly shown in Fig 1.

This is important and very informative data, and is now shown for E17.5, Day 1, and Day 7 thymocytes, as a new figure, Fig. 3. We also display ROR γ t versus CD73 expression within each V γ chain subset, to show the developmental progression of each population (V γ 4, V γ 6, V γ 1/5) through fetal and neonatal ontogeny.

At the end of the paper, there is another aspect that needs additional clarification, which has to do with how “HEB is critical for gdT17 generation through multiple developmental pathways” (lines 332-3). As the authors discuss (lines 327-332), HEB seems to control: the development of fetal CD24⁻ CD73⁻ ROR γ t⁺ Vg6⁺ thymocytes; a later fetal CD73⁺ ROR γ t⁺ thymocyte subset; fetal Vg4⁺ and adult ROR γ t⁺ Vg4⁺ thymocytes. What unifies these distinct pathways in their HEB dependence? On the other hand, what are the HEB-independent gdT17 pathways and (how) do they rely on ROR γ t and Sox4/13 function? I think the authors should further clarify their proposed model on gdT17 development for the benefit of the reader.

The new fetal and neonatal WT data that we collected since the last submission (shown in Figures 1 and 3) was instrumental in helping us to better understand the natural pathways of $\gamma\delta$ T-cell development during ontogeny, and we have used this new understanding to re-structure the narrative of the paper, using the terms Pathway 1 (CD73 upregulation prior to downregulation of CD24) and Pathway 2 (no overt CD73 upregulation prior to CD24 downregulation). Our data clearly shown that Pathway 1 is taken by V γ 5 cells in the fetal thymus (in both WT and HEB^{-/-} mice), and by the majority of adult $\gamma\delta$ T cells (both V γ 1 and V γ 4), whereas Pathway 2 is restricted to V γ 6 fetal thymocytes and V γ 4 neonatal thymocytes. The impact of HEB on Pathway 2 towards the $\gamma\delta$ T17 fate unifies these defects. The discussion has been largely re-written to clarify these connections. We have also added a section on the involvement of potential signaling inputs into each pathway. In particular, we were delighted to read the new manuscript on TCR-mediated ERK signaling as key to suppressing $\gamma\delta$ T17 development (Sumaria et al, Cell Rep. 2017 Jun 20;19(12):2469-2476), which came out after our last submission of our manuscript. This paper contained many of the experiments suggested by the reviewers in response to our first submission, and we have integrated those findings into our conclusions.

Although Il17a PCR data are shown (Fig 1d), i.c.IL-17A (versus IFN-gamma) and ROR γ t protein staining in E17.5 ex vivo CD24- CD73- gd thymocytes should be added in order to define them as bonafide gdT17 cells ex vivo.

This is now shown in Fig. 1c-f (ROR γ t) and in a new Supplementary Fig. 1 (IL-17).

Have the authors analyzed CD24- CD73- gd thymocytes in newborn or young mice? How does this subset behave throughout ontogeny (ex vivo)? Only E17.5 and 12 w.o. thymi are shown, thus leaving a huge gap in between

A more extensive time course of ex vivo thymocyte analysis (E17.5, d1, d7, 4 wks, 7 wks, 12 wks) is now shown in Fig 1b. Earlier FTOC time points are also shown (Fig. 1a).

Reviewer #2 (Remarks to the Author):

This remains a well performed study that demonstrates the complex role of HEB in $\gamma\delta$ T cell development in the thymus.

We thank the reviewer for the supportive comments.

In this revision, the authors have turned away from the original thrust of the paper, which was not well addressed by the data presented, that HEB is a key downstream mediator that integrates TCR signaling to regulate cell fate decisions. Thus, although the study very convincingly shows that HEB is critical for aspects of $\gamma\delta$ T17 development, and demonstrates fetal and adult differences, there is somewhat reduced impact compared to the original goal of this study.

As mentioned in our response to Reviewer #1, we were fortunate enough to be able to read the manuscript on TCR-mediated ERK signaling as key to suppressing $\gamma\delta$ T17 development (Sumaria et al, Cell Rep. 2017 Jun 20;19(12):2469-2476) before finishing this revision, as that paper provides strong evidence for the link that we originally proposed. We have now integrated those findings into our discussion.

The biggest limitation is not knowing how the balance of E proteins and inhibitors leads to functional outcome as proposed. If E protein activity is upstream of Sox4 and Sox13, then E proteins are active in the cells with the highest Id3 (as well as HEB), again leading to the difficulty in inferring overall E protein activity without some actual measure.

We agree that it is difficult if not impossible to deconvolve the molecular interactions within the CD24⁺CD73⁻ population as a whole; single cell resolution will be necessary. Our intracellular flow cytometry data (Figure 1c-e) provides direct evidence that ROR γ t expression is heterogeneous in the CD24⁺CD73⁻ population, regardless of V γ chain usage. This heterogeneity resolves as cells move into Pathway 1 or 2, in a V γ chain specific manner. Therefore, we feel that it is unlikely that HEB, Id3, Sox13, and Sox4 are all expressed concordantly in the same individual cells. Instead, we propose that the CD24⁺CD73⁻ population represents a heterogeneous mixture of precursors receiving different sets of signaling cues (TCR, cytokines) that change over time due to the maturing thymic microenvironment, and that differ according to the identity of the TCR. Accordingly, we have shifted the discussion away from the “balance of Id and HEB proteins” that we originally proposed. ROR γ t and Id3 reporter mice will be useful tools in the future for understanding the interplay between these factors.

Overall, the authors have shored up the original findings and provided some new data, most notably that SOX4 and possibly SOX13 are likely direct targets for regulation by HEB/ E proteins and, consistent with that, that ectopic expression of Id3 can repress SOX4 and SOX13 expression. An additional key finding is that the CD24⁺CD73⁻ cell population contains precursors for CD24⁺CD73⁻ $\gamma\delta$ T17 cells. These are significant results based on technically challenging experiments, and greatly enhance the paper.

We thank the reviewer for the positive comments.

Failure to look in the gut in cKO mice remains a weakness.

We have now evaluated the presence of CD24⁺CD73⁻ $\gamma\delta$ T-cells in the gut, and found that they are not present, shown in Figure 1g.

The reviewer assumes that the authors are talking about the TCF-1 protein, which is encoded by Tcf7, not Tcf1 (line 343).

The discussion of TCF1 (and Tcf7) has been removed.

Error bars are missing from Fig. 1d, Il17a. Fig. 4b, Id3- is n.s. correct for first 2 cell populations?

The data that was contained in Fig. 1d is now shown in Supplementary Fig. 3. It is stated in the figure legend that all comparisons that are statistically significant are indicated, and the others are non-significant, including IL-17. Likewise, Id3 expression is not significant in what is now Fig. 5b. In both cases, there is considerable variability, which we were not able to resolve over many experiments, perhaps due to the imprecision of age of harvest that is inherent in timed matings.

Periods should replace commas in numbers on Figures (?)

This is apparently the default output of the current version of Flow Jo. All commas have now been replaced with periods.

Page 10, reference should be to Supp. Fig. 4 (not 3d)

References to all figures and supplementary figures have been updated and carefully checked.

REVIEWERS' COMMENTS:

Reviewer #1 (Remarks to the Author):

The authors have provided new interesting data that addressed my previous concerns; and greatly improved the discussion of the manuscript, which is now suitable for publication in Nature Communications.

A small detail: when editing the final version of the paper, I suggest replacing " $\gamma\delta$ T17s" by " $\gamma\delta$ T17 cells".

Reviewer #2 (Remarks to the Author):

The authors have largely addressed the previous concerns raised, the use of pathway 1 and 2 is a help to the reader, and the study is a significant addition to understanding the complexity of $\gamma\delta$ effector T cell generation in the thymus. The only comment is that the key seems to be what is happening at (or just prior to) the CD24⁺73⁻ stage, as progenitor heterogeneity appears to already be established there, and this is the subset with highest HEB. The investigators are correct that the next important step is using reporters (and scRNA-seq) to get at these issues, but it is not unreasonable to conclude that that is outside the scope of the current manuscript. However, it is a bit surprising that there is no significant difference in Rorc at this stage in the absence of HEB (no change in Id3 fits with a distinct cell subset). And it is possible that decreases in Sox4/13 affect the maintenance rather than initial upregulation of Rorc, so it would be hard to argue that the data is inconsistent with the overall model. Given the complexity, a final graphical model would also be a welcome addition.

Response to Reviewers
NCOMMS-16-21340B

Reviewer #1 (Remarks to the Author):

The authors have provided new interesting data that addressed my previous concerns; and greatly improved the discussion of the manuscript, which is now suitable for publication in Nature Communications.

A small detail: when editing the final version of the paper, I suggest replacing " $\gamma\delta T17s$ " by " $\gamma\delta T17$ cells".

We have made the suggested edit from $\gamma\delta T17s$ to $\gamma\delta T17$ cells throughout the manuscript.

Reviewer #2 (Remarks to the Author):

*The authors have largely addressed the previous concerns raised, the use of pathway 1 and 2 is a help to the reader, and the study is a significant addition to understanding the complexity of $\gamma\delta$ effector T cell generation in the thymus. The only comment is that the key seems to be what is happening at (or just prior to) the CD24+73- stage, as progenitor heterogeneity appears to already be established there, and this is the subset with highest HEB. The investigators are correct that the next important step is using reporters (and scRNA-seq) to get at these issues, but it is not unreasonable to conclude that that is outside the scope of the current manuscript. However, it is a bit surprising that there is no significant difference in *Rorc* at this stage in the absence of HEB (no change in *Id3* fits with a distinct cell subset). And it is possible that decreases in *Sox4/13* affect the maintenance rather than initial upregulation of *Rorc*, so it would be hard to argue that the data is inconsistent with the overall model. Given the complexity, a final graphical model would also be a welcome addition.*

We thank the reviewer for the positive comments and share the enthusiasm for progressing on to the next steps after the current paper is published. We have now added a **Figure 9**, which presents a graphical model that integrates many aspects of our findings including developmental pathways, CD24/CD73 expression, V γ chain usage,

timing during fetal/neonatal/adult development, TCR/ERK signaling, Id3/HEB opposition, and HEB control of Sox genes. We think this is a great addition that will help our readers understand the model we have generated to explain gdT17 pathways of development.